# Wild or Reared? *Cassiopea andromeda* Jellyfish as a Potential Biofactory

**DOI:** 10.3390/md23010019

**Published:** 2025-01-01

**Authors:** Stefania De Domenico, Andrea Toso, Gianluca De Rinaldis, Marta Mammone, Lara M. Fumarola, Stefano Piraino, Antonella Leone

**Affiliations:** 1Consiglio Nazionale delle Ricerche, Istituto di Scienze delle Produzioni Alimentari, (CNR-ISPA)–Lecce, Via Monteroni, 73100 Lecce, Italy; stefania.dedomenico@ispa.cnr.it (S.D.D.); gianluca.derinaldis@ispa.cnr.it (G.D.R.); 2Dipartimento di Scienze e Tecnologie Biologiche ed Ambientali (DiSTeBA), Università del Salento, Via Monteroni, 73100 Lecce, Italy; andrea.toso@unisalento.it (A.T.); marta.mammone@unisalento.it (M.M.); laramarastella.fumarola@unisalento.it (L.M.F.); stefano.piraino@unisalento.it (S.P.); 3Consorzio Nazionale Interuniversitario per le Scienze del Mare (CoNISMa), Local Unit of Lecce, Via Monteroni, 73100 Lecce, Italy; 4ENEA Research Centre of Brindisi, Department of Sustainability, Circularity and Climate Change Adaption of Production and Territorial Systems, SS 7 Appia Km 706, 72100 Brindisi, Italy; 5NBFC, National Biodiversity Future Center, 90133 Palermo, Italy

**Keywords:** *Cassiopea andromeda*, farming, holobiont, zooxanthellae, pigments, fatty acids, antioxidant, sustainable production

## Abstract

The zooxanthellate jellyfish *Cassiopea andromeda* (Forsskål, 1775), a Lessepsian species increasingly common in the western and central Mediterranean Sea, was investigated here to assess its potential as a source of bioactive compounds from medusa specimens both collected in the wild (the harbor of Palermo, NW Sicily) and reared under laboratory-controlled conditions. A standardized extraction protocol was used to analyze the biochemical composition of the two sampled populations in terms of protein, lipid, and pigment contents, as well as for their relative concentrations of dinoflagellate symbionts. The total extracts and their fractions were also biochemically characterized and analyzed for their in vitro antioxidant activity to quantify differences in functional compounds between wild and reared jellyfish. The two populations were similar in terms of extract yield, but with substantial differences in biomass, the number of zooxanthellae, protein and lipid contents, and fatty acid composition. The hydroalcoholic extracts obtained from jellyfish grown under controlled conditions showed greater antioxidant activity due to the presence of a higher content of bioactive compounds compared to wild jellyfish. This study could be the basis for considering the sustainable breeding of this holobiont or other similar organisms as a source of valuable compounds that can be used in the food, nutraceutical, or pharmaceutical sectors.

## 1. Introduction

Since the Japan Zoo and Aquarium began to show the jellyfish *Aurelia aurita* [1] in 1967, the interest in the breeding of jellyfish has grown exponentially over time due to the potential of jellyfish in a wide range of uses and research areas. Besides their common use as marine ornamental species, jellyfish are considered an important resource in the traditional Asian food industry [2,3] as feed for other organisms, and they could serve as raw material in a broad array of biotechnological applications [4] associated with the biomaterial [5,6], pharmaceutical [3,7,8,9], cosmeceutical [10,11], nutraceutical, and food industries [8,12,13,14,15,16,17,18,19,20,21,22,23].

Scyphomedusae contain a large volume of water (>95%) and possess some compounds that may have biotechnological applications, such as collagen, fatty acids, and bioactive compounds extracted from the crude venom of their nematocysts. Zooxanthellate jellyfish are also enriched with biomass and specific bioactive compounds from endosymbiont microalgae. Therefore, structural proteins such as collagen have garnered significant attention for their versatile applications in biomedicine, nutrition, and as nutraceuticals. Collagen can be enzymatically hydrolyzed to produce bioactive peptides with antioxidant and anti-inflammatory properties which contribute to its health-promoting effects, particularly in managing oxidative stress- and inflammation-related conditions. Additionally, jellyfish collagen’s physicochemical properties—such as its high biocompatibility, biodegradability, and mechanical strength—make it a highly valuable biomaterial in tissue engineering, wound healing, and drug delivery systems. Lipids, particularly unsaturated fatty acids, provide significant health benefits related to the prevention of cardiovascular diseases. In addition, bioactive compounds such as phenols and carotenoids, primarily derived from symbiotic organisms (e.g., algae in symbiotic relationships), exhibit strong antioxidant properties. These antioxidants play a critical role in neutralizing free radicals and reducing oxidative stress, which are key contributors to chronic diseases with an oxy-inflammatory basis, such as atherosclerosis, type 2 diabetes, and certain cancers.

Recently, we focused our research on the jellyfish *Cassiopea andromeda* (Forsskål, 1775), a non-native Mediterranean species, to corroborate the hypothesis of its high potential as a source of molecules with various biological activities and biotechnological applications. *C. andromeda* could indeed be an important source of both proteins and proteinaceous hydrolysates [24] with antioxidant activity, as already demonstrated for other scyphozoan jellyfish, like *Rhizostoma pulmo* [20], *Rhizostoma luteum* [17], *Aurelia coerulea* [12,25], and *Nemopilema nomurai* [26]. As a holobiome hosting symbiotic photosynthetic dinoflagellates (family Symbiodiniaceae), known as zooxanthellae, *C. andromeda* is also a source of other valuable compounds easily obtainable by hydroalcoholic extraction [15,24]. Indeed, additional, yet uncharacterized molecules may be extracted from this gelatinous species, potentially due to physiological interactions with autotrophic microorganisms [27]. The mutualistic relationship with zooxanthellae is common in several invertebrate hosts (protists, sponges, cnidarians, and mollusks): by their association, dinoflagellates [28,29] benefit from the inorganic nutrients (CO_2_, NH_3_ and PO_4_^3−^) excreted by the animal host, and in turn the metazoan is rewarded by the photosynthates produced by the endosymbiotic protists [30]. This symbiotic relationship significantly influences the production of bioactive compounds, including products of both primary metabolism (e.g., proteins and lipids) and specialized secondary metabolism. Examples include photosynthetic metabolites produced by zooxanthellae, such as antioxidants and UV-absorbing compounds that protect both the host and the symbiont from oxidative stress, as well as photosynthetic products used as precursors by the host to form derived compounds involved in defense against predators and pathogens. Finally, the symbiotic interaction may upregulate the production of unique bioactive compounds under environmental stress (e.g., temperature or UV irradiation).

Although the majority of studies on cnidarian photosymbiosis are focused on scleractinian corals, the genus *Cassiopea* is the most studied among mixotrophic jellyfish [31,32,33,34,35], with up to 12 known species, all generally known as upside-down jellyfish due to their epibenthic life-style. Most of the time, these jellyfish lay on the sea floor on their exumbrellar side, with the oral arms and mouth pointed towards the water surface to improve the photosynthesis of zooxanthellae, which reside in the subumbrella endoderm. *Cassiopea andromeda* lives in shallow tropical and sub-tropical waters. Native to the Red Sea, it reached the Mediterranean Sea shortly after the opening of the Suez Canal. Since then, it has spread across the coastal waters of Malta, Tunisia, Spain, and Italy (Sicily) [36,37,38,39,40]. From the umbrella to the oral arms, *Cladocopium*, *Symbiodinium*, and *Breviolum* spp. are its most common endosymbionts [41,42]. In particular, *Symbiodinium* species and their photosynthetic machinery have been studied for their importance in the massive bleaching events of zooxanthellate corals observed in the last three decades. The photosynthetic machinery is known as a source of active compounds and has been regularly investigated to deepen our understanding of the biochemical production of essential nutrients (i.e., polyunsaturated fatty acids), secondary metabolites, and other unique bioactive molecules [43].

Pigment composition in photosynthetic dinoflagellates is distinct from that of any other microalgae because it contains a particular carotenoid (peridinin) as a component of light-harvesting complexes (LCHs) and a protector of cells from oxidative stress thanks to a unique photosynthetic antenna system including peridinin/chlorophyll-a/proteins (PCPs) [44,45,46]. HPLC analysis carried out on *Symbiodinium* isolated from corals showed the dinoflagellates contain two types of chlorophyll (Chl-c2 and Chl-a), b-carotene and phaeophytin-a, and the xanthophylls peridinin, diadinoxanthin, diatoxanthin, and diadinochrome [47,48]. When analyzed in cultures of *Symbiodinium* strain Y106, the main pigment was chlorophyll-a, followed by peridinin (representing about 60% of the total carotenoids of the cells) and then by diadinoxanthin, b-carotene, and others [49].

In coral–microalgae symbiosis, zooxanthellae convert inorganic carbon and nitrogen into organic photosynthetic products to translocate some of these to the host, mainly in the form of carbohydrates, amino acids, and fatty acids. The latter are known as a combined source of metabolic energy, essential nutrients, and omega-3 and omega-6 polyunsaturated fatty acids (PUFAs), as well as precursors of bioactive metabolites [50].

Temperature and light affect the metabolism of pigments, lipids, and fatty acids and consequently the biochemical composition and content of bioactive compounds of *Symbiodinium* cells and their host animal tissues [51,52]. For example, increased light intensity can inhibit algal growth [53] or modify the composition of polyunsaturated fatty acids (PUFA) in dinoflagellates [54]. The production and translocation of fatty acids by symbiotic zooxanthellae to the host tissues was studied also in the jellyfish *Cassiopea* sp., where a decrease in light intensity caused a reduction in valuable polyunsaturated fatty acids, such as 18:1ω9 and the abundant 18:4ω3, in the microalgae and their concurrent increase in the jellyfish tissues, suggesting a possible transfer from the symbionts to the host [32].

Compounds such as long-chain fatty acids and PUFAs, as well as other molecules of plant origin (photosynthetic pigments, phenolic compounds), have been shown to be potentially bioactive molecules. Carotenoids are natural plant compounds with antioxidant and anti-carcinogenic activities [55]; in addition to the vast literature on the biological activities of plant molecules, polyunsaturated and monounsaturated fatty acids from Antarctic algae have the potential to reduce the proliferation of and induce apoptosis in breast cancer cells [56]. Further, peridinin exerts anti-proliferative and pro-apoptotic effects in vitro by suppressing NF-κB and Akt signaling in HTLV-1-infected T cells [57], and the algal PCP complex exhibits antioxidant, anti-inflammatory, and anti-cancer activity against human metastatic breast adenocarcinoma (MDA-MB-231) cancer cells [58].

Marine organisms have been considered a promising resource for biotechnological applications for many years [59,60], but their wide potential is still largely unexplored. In this respect, the development of efficient cell factories stands out as one of the major bottlenecks for the further advancement of industrial biotechnology [61]. Marine ecosystems, rich in unexplored habitats and organisms, can serve as an ideal benchmark for the development of new integrative and systems biology approaches and provide opportunities for the development of efficient biofactories containing significant potential to overcome the challenges of biotechnological applications. Some specific challenges of marine biotechnology include advancing (a) techniques for cultivating marine organisms and their isolated cells through reliable protocols and profitable production schemes; (b) research on genetics of marine species; (c) knowledge on symbiotic relationships, the biology and chemistry of defense mechanisms, and the chemo-ecological network underlying marine invasions; and (d) strategies found in prokaryotes to adapt to extreme environments.

The biotechnological potential of symbiotic microalgae is renowned, as dinoflagellates are a valuable source of unique secondary metabolites and pigments of interest for biomedical, nutraceutical, pharmaceutical, and cosmeceutical applications. However, scaling the processes involved in cell culture remains a significant challenge; at the same time, the complete spectrum of secondary metabolites produced by algal cells cannot be generated without their animal host.

In a previous work [15], we investigated the occurrence of zooxanthellae in *C. andromeda* jellyfish from the Palermo harbor (Italy). This species entered the Mediterranean Sea through the Suez Canal more than 100 years ago [62] but it was detected in Sicily for the first time in 2014 [39].

Separately for tissues of the umbrella and oral arms of the jellyfish, we analyzed the relative composition of proteins, fatty acids, phenol compounds, and pigments, as well as the relative antioxidant activities, in hydroalcoholic extracts and in lipophilic and hydrophilic fractions obtained from both the umbrella and oral arms [15].

Wild jellyfish can be exploited as a marine resource for different uses [2,3,63]; however, their presence and quantity are considered unpredictable due to the seasonality and spatiotemporal variability of their life cycles [64].

To overcome fluctuations in jellyfish availability, production through aquaculture could offer a constant supply stock of jellyfish biomass, free from unknown pollutants, with assured traceability and sustainability of exploitation. Of course, the cultivation of non-native species also poses several challenges and serious risks that must be carefully considered before considering the benefits of economic growth, increased food security, and possible ecological services. Environmental risks due to invasive potential, predation, and competition with native species, the presence of pathogens, as well as regulatory and ethical issues related to the alteration of natural ecosystems for the benefit of humans in the context of the climate crisis, must be seriously considered.

In this work, we compare the biomass of *C. andromeda* jellyfish reared in an aquarium under controlled conditions and that of wild jellyfish sampled in the harbor of Palermo for their potential as a source of proteins and bioactive molecules. Their whole biomass, hydroalcoholic extracts and related fractions, and the composition of chlorophylls, carotenoids, and essential fatty acids were analyzed and compared. The protocol described by Leone et al. [8,24], already used on wild samples of *C. andromeda* [15], was applied to reared jellyfish to evaluate and compare their possible uses as sources of bioactive molecules for applications of interest in the pharmaceutical, cosmeceutical, and biomedical, sectors.

## 2. Results and Discussion

Biochemical analyses were carried out on juvenile (3-month-old) specimens of *C. andromeda* jellyfish (also referred herein as holobionts based on their symbiotic association with zooxanthellae), collected either from the wild in the port of Palermo, Italy (wild), or from a laboratory-based population (reared) established at the University of Salento, Italy (Figure 1). Whole-jellyfish (WJ) biomass of wild and reared holobionts was analyzed and compared in terms of its protein, lipid, and pigment content, fatty acid composition, as well as the number of zooxanthellae and their taxonomic identity. Further, using an established extraction protocol [8,15,24], hydroalcoholic jellyfish extracts (ExDW) and their fractions were analyzed for protein, lipid, fatty acid composition, total phenolic content, and in vitro antioxidant activity to evaluate potential biochemical differences in extracts obtained from the two jellyfish populations (wild vs. reared).

### 2.1. Holobiont Biomass Characterization

#### 2.1.1. Biometric Data

Biometric measurements such as the fresh weight (FW) and umbrella (UMB) and oral arm (OA) diameter of *C. andromeda* holobionts were taken shortly after collection from representative specimens from the wild (Figure 1A) and from the laboratory-reared populations (Figure 1B).

The biometric data are listed in Table 1. Overall, 3-month-old wild specimens had a statistically larger diameter than individuals of the same age cultured under controlled conditions, both for the umbrella (UMB) and the oral arms (OAs), as well as for the fresh weight (FW) and the FW/diameter ratio. The wild jellyfish also had higher values for dry weight (DW), DW/diameter ratio, and DW expressed as FW percentage than the reared specimens. After freeze-drying, the average DW of *C. andromeda* was 5.7% of FW for wild specimens and 4.5% of FW for the laboratory-reared specimens. The biometric data suggest that size and consistency, intended as the accumulation of dry biomass per fresh weight, are significantly greater in jellyfish grown in the wild compared to those cultured in aquarium. These differences are certainly due to differences in environmental factors, particularly the availability of certain nutrients. However, targeted experiments are still required to identify the specific parameters involved.

#### 2.1.2. Symbiont Identification and Quantification in *Cassiopea andromeda* Specimens

As other cnidarians, members of the upside-down jellyfish *Cassiopea* genus are mixotrophic, obtaining energy both from heterotrophic feeding and from photosynthates produced by their unicellular symbionts, i.e., several species of dinoflagellates (also known as zooxanthellae) belonging to the family Symbiodiniaceae. The abundance and identity of symbiotic zooxanthellae in the cnidarian gastrodermis may significantly influence the life cycle and photo-physiology of the host, as well as the overall holobiont biochemical composition. To identify the zooxanthellae hosted in the jellyfish studied here, a molecular barcoding analysis was performed on symbiont DNA extracted from five jellyfish collected at the port of Palermo, as well as from laboratory-reared polyp populations of *Cassiopea*, through the amplification and sequencing of the symbiont chloroplast’s large ribosomal subunit (cp23S-rDNA sequencing). For the laboratory-reared population, DNA barcoding of zooxanthellae was carried out on the polyp stage, because dinoflagellates from the environmental pools are known to enter the host tissues in the polyp stage only; later, they are vertically transmitted to the ephyra during the strobilation process [42].

The presence of zooxanthellae in polyp and medusa tissues was confirmed by microscopic observation (Appendix A).

Molecular analysis results showed that wild jellyfish hosted symbionts of different genera but of the same family Symbiodiniaceae. One wild jellyfish specimen (CAS_W2) hosted *Symbiodinium necroappetens* (previously named clade A13, [41]); CAS_W5 hosted *Breviolum pseudominutum* (former clade B1); meanwhile, in the last three samples (CAS_W1, CAS_W3, and CAS_W4) the species was not identified, but only the genus (*Symbiodinium* sp.). Our results are in line with what previously observed in the literature, where *Cassiopea* sp. was commonly found in symbiosis with dinoflagellates of the genus *Symbiodinium* [31,42]. Also, the polyps grown in the aquarium hosted *S. necroappetens* (clade A13), suggesting the presence of this zooxanthellae species in the seawater used in aquarium management. Actually, polyps acquire the clade available in the water, and they have a higher chance of survival and of having strobilation if they acquire clade A or B than other clades [65].

In addition, the number of symbionts in whole jellyfish (WJ) were measured per gram of dry (DW) and fresh weight (FW) biomass (microalgae/g WJ-DW and microalgae/g WJ-FW, Table 2). The specimens of jellyfish grown in aquarium were pooled in three samples (Cas_R1, Cas_R2, and Cas_R3). The wild-caught specimens showed some variability in the density of zooxanthellae hosted in their tissues, whereas it was more homogeneous in reared jellyfish tissues (Table 2). On average, the number of symbionts in wild individuals was 60.6 × 10^6^ microalgae /g WJ-DW and 3.36 × 10^6^/g WJ-FW, while in the biomass of reared jellyfish, there was about 23.9 × 10^6^/g WJ-DW corresponding to 1.07 × 10^6^/g WJ-FW, about a third compared to jellyfish found in the free marine environment.

### 2.2. Biochemical Characterization

#### 2.2.1. Hydroalcoholic Extraction

In our previous works, we demonstrated that *Cassiopea andromeda* jellyfish with its endosymbionts (holobiont) could be a source of biologically active and easily extractable compounds, such as chlorophylls, pigments, and lipids [15,24]. Based on these protocols, the biomass of wild and reared *C. andromeda* holobionts was characterized and compared by hydroalcoholic extractions performed on *C. andromeda* jellyfish from natural environment (Palermo harbor) or grown in controlled conditions (aquarium). Hydroalcoholic-soluble compounds (lipids, phenols, pigments, and soluble proteins) were extracted from each sample with 80% ethanol solution. The amount of the total extract (ExDW), expressed as grams of dry weight and as percentage of the lyophilized samples (yield%), were measured for each jellyfish sample. The extraction yields (expressed as % of DW of total holobiont) were statistically comparable between the wild and reared jellyfish, at 46.2 ± 9.6% and 40.6 ± 7.2% of extract per lyophilized wild and reared jellyfish specimens, respectively (Appendix A).

#### 2.2.2. Soluble and Insoluble Protein Content

Jellyfish components insoluble in 80% ethanol, consisting mainly of proteins, including collagen, were subjected to sequential enzymatic digestion with pepsin and collagenase, as reported by De Domenico et al. [20]. The soluble protein content present in ExDW and in pepsin and collagenase-hydrolyzed fractions were quantified by colorimetric analysis and expressed as mg/g of WJ-DW (Table 3). No differences were evident between the soluble proteins obtained by hydroalcoholic extraction from either wild or reared jellyfish specimens (about 5 mg/g WJ-DW), while high variability was observed in the total protein content of samples grown in a natural environment compared to reared holobionts (42.44 ± 20.26 mg/g WJ-DW and 26.20 ± 7.80 mg/g WJ-DW, respectively). The differences were mainly due to the insoluble fraction, in particular the pepsin-digestible proteins. The amount of collagenase-hydrolyzable protein expressed per gram of total dry weight was not significantly different between wild and reared holobionts. Approximately 20% and 10% undigested protein material was measured in wild and reared jellyfish, respectively. Since collagen, a structural protein of the host jellyfish, does not appear to vary much in the two holobiont populations (Table 3), it is possible to infer that the greater variability is due to proteins originating from the symbionts and/or from the symbiont–host metabolic interactions in the two different environmental conditions. Indeed, the difference in the number of symbionts between wild and farmed samples overlaps with the difference in the content of non-collagenous fibrillar proteins, i.e., pepsin hydrolyzed proteins, which in reared holobionts are about a third of those in wild specimens. Therefore, the number of zooxanthellae per dry weight could become a parameter to be considered to predict and/or increase the total protein content of the holobiont.

In Figure 2, the SDS-PAGE analysis of proteins present in samples representative of whole jellyfish (WJ), total hydroalcoholic extracts (ExDW), and pepsin- and collagenase-hydrolyzed protein fractions (P and C, respectively) is shown. After electrophoretic separation, polypeptides were visualized by both a stain-free system (Figure 2A) and standard Coomassie Brilliant Blue (CBB) staining (Figure 2B) to analyze any differences between the two patterns due to the presence of proteins with low tryptophan content, such as collagen, which are better highlighted by CBB staining, as already reported in De Domenico et al. [20].

Figure 2 shows that the total proteins of whole jellyfish (WJ) of both wild and farmed JF samples consist of many polypeptides over a wide molecular weight (MW) range. Of these, only some low-molecular-weight (<20 kDa) polypeptides are soluble in 80% ethanol and detectable by SDS-PAGE (ExDW). Several qualitative differences are detectable between the wild and reared JF polypeptide patterns, clearly evident at low MW ranges (15–20 kDa). Wild JF show two bands of about 16 and 18 kDa in the total protein WJ line, also present in the related ExDW (Figure 2B), while reared JF protein profiles show three bands (about 15, 16, and 18 kDa), of which only one (18 kDa) was found in the hydroalcoholic extract (ExDW line). Extractable proteins could contain the light harvesting complex (LHC) peridinin/chlorophyll a/protein (PCP), a soluble protein complex characteristic of photosynthetic dinoflagellates. Dinoflagellate PCPs show great variation in amino acid sequences and spectroscopic properties and can occur in two types—a homodimer of 14–16 kDa monomeric units and a monomer form of 30–35 kDa molecular mass, possibly due to gene duplication [45,58]. In addition to water-soluble PCPs, thylakoid intrinsic trimeric chlorophyll a/c2/peridinin/protein complexes (acpPCs) of 18–20 kDa have also shown abundance in *Symbiodinium* cells (58). The polypeptide profiles of the wild and reared analyzed jellyfish populations detected by SDS-PAGE evidence that they could have different LHC protein compositions, with the PCP homodimeric form present only in reared jellyfish, perhaps due to the artificial lighting applied.

Although further studies are needed to demonstrate a structural and molecular difference in PCPs in jellyfish grown in the two environments (wild and farmed), it would be interesting to consider the differences induced by different types of lighting, artificial and natural, on the composition of the LHCs.

SDS-PAGE polypeptide profiles of pepsin- and collagenase-hydrolyzed proteins from both wild and reared jellyfish are visualized in lines P and C, respectively. The two samples showed somewhat similar peptide patterns, with some quantitative–qualitative differences. Both the pepsin- and collagenase-hydrolyzed fractions of wild JF are richer in peptides than those derived from reared samples. In particular, the pepsin-hydrolyzed fraction (line P) shows a number of high-molecular-weight peptides (>100 kDa) and a band at approximately 17 kDa, but two bands at approximately 30 and 21 kDa are missing (evident in the P line of reared JF, Figure 2A, and confirmed by Figure 2B).

#### 2.2.3. Identification and Quantification of Microalgal Pigments in Hydroalcoholic Extracts

To compare the endosymbiotic zooxanthellae pigments present in wild and reared *C. andromeda* jellyfish, the identification and quantification of the main characteristic pigments derived from microalgae present in *C. andromeda* were performed by HPLC analysis using appropriate standards for the pigment characteristics of *Symbiodinium* sp. (chlorophyll-a and the carotenoids b-carotene, peridinin, and diadinoxanthin) and other photosynthetic microorganisms (lutein). The results summarized in Table 4 show the amounts of pigments expressed as mg/g of total dry JF biomass (WJ-DW), as mg/g of hydroalcoholic extract (ExDW), and as pg per number of microalgae to eliminate any bias caused by the compositional difference and number of microalgae between wild and reared jellyfish. All the five analyzed pigments were found in both wild and reared jellyfish. The most abundant pigment in wild jellyfish was chlorophyll-a (1327.4 ± 483.1 mg/g ExDW), followed by lutein (776.7 ± 328.3 mg/g ExDW), peridinin isomers (136.2 ± 94.4, 19.6 ± 8.6, and 26.2 ± 12.5 mg/g ExDW), diadinoxanthin (82.2 ± 54.9 mg/g ExDW), and b-carotene (6.6 ± 1.9 mg/g ExDW).

A similar concentration pattern was detected in jellyfish grown in the aquarium for peridinin isomers (86.1 ± 19.4, 20.8 ± 3.3, and 18.6 ± 4.4 mg/g ExDW), diadinoxanthin (68.0 ± 10.5 mg/g ExDW), and b-carotene (4.8 ± 0.6 mg/g ExDW). Nevertheless, a statistically lower concentration of chlorophyll-a (598.9 ± 128.2 mg/g ExDW) and lutein (167.1 ± 47.5 mg/g ExDW) in reared as compared to wild jellyfish was found. Similar ratios were found when pigment concentrations were related to the dry biomass of the jellyfish and expressed as mg/g WJ-DW (Table 4), also proving the good extraction efficiency of our protocol.

With respect to peridinin quantification, we were able to distinguish diverse peaks relative to peridinin isomers from the HPLC chromatograms (Appendix A), as already described [48,49,66]. The occurrence of lutein could indicate the presence of other types of photosynthetic organisms (such as microscopic endolithic algae) present in the samples by accidental contamination [67] or the presence of photosynthetic bacteria in the gastric cavity of *C. andromeda* as reported by Viver [68] in *C. tuberculata*. The different content of lutein between wild and reared holobionts was also observed when the xanthophyll amounts were expressed as the number of microalgae present in the samples (as pg of pigment/microalgae, Table 4). This therefore suggests either an actual decrease in lutein content due to the stress of the non-natural environment, such as the applied artificial light and the concentration of nitrogen, which influence its biosynthesis, or the presence of a greater number of different photosynthetic microorganisms in wild samples. The quantity of chlorophyll-a, specific to *Symbiodinium* sp., is the same in the two populations when expressed in picograms of pigment/microalgae. The lower amount of chlorophyll-a in reared jellyfish as compared to the wild specimens is therefore due to a lower presence of Symbiodiniaceae in their biomasses.

Comparing the carotenoid/chlorophyl ratios, differences between the ratio values of wild and reared jellyfish samples were evident. Indeed, the ratio between the sum of all tested carotenoids and chlorophylls was higher (about 0.8) in wild specimens than reared holobionts (about 0.6). In particular, in reared holobionts, the b-carotene and diadinoxanthin/chlorophyll ratios were almost double those of wild holobionts, and the peridinin/chlorophyll ratio was more than a third higher, whereas the lutein/chlorophyll ratio in reared holobionts was half that in wild holobionts. This could indicate a different composition of the microbiome and/or their different metabolism in wild and reared jellyfish in response to stress due to the rearing conditions (lighting, temperature, nutrients, etc.). This would be worth investigating in subsequent and more specific experiments.

### 2.3. Fractionation and Characterization of Hydroalcoholic Soluble Compounds 

Jellyfish hydroalcoholic extracts (ExDW) containing a mixture of heterogeneous compounds, including pigments, were subjected to fractionation with acetonitrile/water solution (1:1 *v*/*v*) to obtain a hydrophilic lower phase (LP) containing soluble proteins and an upper phase (UP) containing hydroalcoholic soluble compounds, as previously reported [8,15,24].

Since LPs and Ups have been demonstrated to be enriched mixtures of biological active compounds [15], both fractions from each *C. andromeda* sample were analyzed for fatty acid compositions, lipid, protein, phenol, and pigment contents and compared with the parent solution of ExDWs. It should be noted that the two phases appeared separated by an intermediate phase (IP), which was semi-solid and therefore difficult to analyze using colorimetric assays, and for which only HPLC analysis was possible.

#### 2.3.1. Lipid Content and Fatty Acid Composition

Total lipids were extracted from the whole biomass (WJ), hydroalcoholic extracts (ExDWs), and the lipophilic upper phase (UP) in order to evaluate the total lipid content and the fatty acid composition of *C. andromeda* growth under controlled conditions (reared) in comparison with our available data on wild specimens [15,24]. A considerable amount of lipids was obtained (Table 5) from the whole biomass of reared *C. andromeda*, namely 605.8 ± 305.0 mg/g WJ-DW, corresponding to about the 60% of the dry biomass. This value was significantly higher than those obtained in our previous experiments from wild individuals (about 9.4 ± 0.4 mg/g WJ-DW) from data in [24], probably due to the continuous and abundant feeding with *Artemia salina* (see Section 3.1, in Material and Methods). Furthermore, extraction with 80% EtOH was able to separate a considerable amount of lipids, equal to 549.3 ± 225.9 mg/g ExDW (about the 90% of WJ-DW), and a further lipid enrichment occurred in the UP during fractionation (3442.3 ± 1160.6 mg/g ExDW). In reared holobionts, no lipids were detected in the hydrophilic fraction LP of the hydroalcoholic extract (Table 5), as also found in the LP of wild jellyfish [15,24].

Table 6 shows the fatty acid (FA) composition of lipids present in whole-jellyfish tissues (WJ), 80% ethanol extract (ExDW), and the upper phase of the hydroalcoholic extract (UP) from reared *C. andromeda* holobionts. Data are expressed as percentages of total FAs. The reared holobionts were richer in saturated FAs (SFAs, about 66.7 ± 9.2% of total lipids) than the wild holobionts previously analyzed [24], which contained about 48% of total lipids (Appendix A). The monounsaturated FA content in reared holobionts (MUFAs, 5.4 ± 1.0% of total lipids) was similar to MUFAs found in wild jellyfish (about 6% of total lipids), while a lower content of polyunsaturated FAs (PUFAs, 27.9 ± 8.4%) was found in reared jellyfish as compared to the wild holobionts (44% of total lipids, Appendix A). A similar pattern was found in the hydroalcoholic extract and in its lipophilic fraction UP (51.2 ± 3.4% and 60.5 ± 2.9% of total lipids, respectively), as also seen in wild jellyfish samples (Appendix A). In reared jellyfish samples, MUFAs were detected at the same concentration in the extract (4.2 ± 0.4% of total lipids) and in the UP (2.0 ± 0.5%), whereas a statistically significantly different enrichment of PUFAs in both ExDW (44.6 ± 2.6% of total lipids) and UP (37.5 ± 3.1% of total lipids) was detected when wild specimens of *C. andromeda* holobionts were analyzed in our previous works [15,24].

In comparison with wild holobionts, the cultured jellyfish displayed a distinct composition in terms of the detected fatty acids and their relative percentages (Appendix A). Lauric acid (C12:0) was not detected in either wild or reared jellyfish, and margaric acid (C17:0, 0.7%) was only detected in WJ and in ExDW but not in UP; it was likely lost during the fractionation process. Comparing the FAs detected in reared holobionts (Table 6) and in wild holobionts, summarized in Appendix A, we can note the following: (i) a complete lack of oleic acid (C18:1 cis-9) and the presence of vaccenic acid (C18:1 trans-11) instead; (ii) a lower concentration of isolinoleic acid (C18:2 cis-6,9) in reared jellyfish samples and its detection in the UP only after enrichment; (iii) the presence of an appreciable percentage of g-Linolenic acid (2.6 ± 0.7%) only in reared jellyfish, but not in wild jellyfish; and (iv) no detection of eicosadienoic acid (C20:2), which was instead found in wild jellyfish, although only after enrichment by UP fractionation [15,24].

The different FA composition in reared holobionts, both in qualitative and quantitative terms, could very likely be due to the daily feeding with *A. salina* nauplii. Indeed, when we extracted and analyzed the lipids of *A. salina* nauplii one day after hatching using the same protocol, they showed an average lipid content of 42.4 ± 15.1% on DW, a higher percentage than the content of 18.9 ± 4.5% on DW reported in the literature [69]. Notably, the FA composition of *A. salina* nauplii (Table 6) showed (i) the presence of about 12% of g-Linolenic acid, detected only in reared jellyfish; (ii) a paucity of essential PUFAs (27.7 ± 0.1%); and (iii) a high content of SFAs (33.6 ± 0.2% of total lipids), mainly palmitic acid (C16:0, 18.1 ± 0.9%), as also reported in the literature [70]. Therefore, although the predominant lipids produced by *Symbiodinium* are palmitic (C16:0) and stearic (C18:0) fatty acids and their unsaturated analogs [71], the highest content of SFAs in reared WJ *C. andromeda* (about 67% of the total FAs) could be due to both feeding with *A. salina* and the lower number of microalgae per DW (see Table 2), which consequently synthesize a smaller quantity of PUFAs.

This proves that FA composition strictly depends on living conditions, including nutrition, paving the way for possible parameter adjustments to regulate the living conditions and shape the lipid composition of farmed *C. andromeda* holobionts.

In fact, it could be assumed that the characteristics of the artificial aquarium environment, mainly the lighting, may have been inappropriate compared to the natural conditions to which the wild *C. andromeda* were subjected, resulting in a decrease in the number of zooxanthellae, also confirmed by the decrease in Chl-a content (Table 4) and a possible prevalence of heterotrophic feeding.

Nevertheless, in Table 6, it is possible to observe an enrichment in PUFA content as a consequence of hydroalcoholic extraction (ExDW) and fractionation in the lipophilic phase (UP), especially in essential fatty acids such as arachidonic acid (C20:4ω6), eicosapentaenoic acid (C20:5ω3), and docosahexaenoic acid (C22:6ω3), whose extracted amounts are comparable to the values obtained from wild jellyfish (Appendix A); meanwhile, the amount of the newly detected g-Linolenic acid (C18:3w6) was preserved. This finding is particularly interesting as such PUFAs have recognized beneficial effects in neurodegenerative and cardiovascular diseases, and also as cancer auxiliary agents for cancer therapy, as highlighted by a plethora of studies [72,73,74,75,76].

#### 2.3.2. Protein and Phenol Content in Hydroalcoholic Extracts and in Their Lipophilic Upper Phase (UP) and Hydrophilic Lower Phase (LP)

The total extract (ExDW), the lipophilic phase (UP) and the hydrophilic phase (LP) from the two populations of *C. andromeda* were also analyzed for their protein (Figure 3A) and total phenolic contents (Figure 3B).

Protein contents, expressed as mg of protein/g ExDW, were similar in the total extracts (ExDW) obtained from the two *C. andromeda* populations, with an amount of 6.2 ± 0.6 and 6.5 ± 0.9 mg/g of ExDW for wild and reared jellyfish, respectively. These data are comparable to those of the parallel experiments shown in Table 3. The differences between the amounts of proteins detectable in the two phases (UP and LP) derived from wild jellyfish were not significant, whereas a different distribution of proteins was found in the two phases obtained from reared jellyfish, with a higher concentration of proteins in the LP than in the UP. Indeed, protein contents were similar in the LPs of reared and wild jellyfish. It is probable that, in reared jellyfish, a large amount of protein could be present as a protein–pigment complex, and that in the fractionation method used here proteins are most likely fractionated in the intermediate phases (IP) and therefore not detectable in LP (personal observation; data not shown). Finally, the higher amount of protein in the UP fractions of wild jellyfish compared to reared ones is likely due to the higher number of zooxanthellae in the wild jellyfish population.

The total phenolic contents in the extracts and the UP and LP were also evaluated and expressed as mg of gallic acid equivalent (GAE) per gram of ExDW (Figure 3B). The total extract from *C. andromeda* grown in the aquarium was richer in phenols than the extract obtained from wild jellyfish. In each sample, the lower phases were richer in phenols than the respective UPs, with a mean concentration of 1793.6 ± 892.5 mg GAE/g of ExDW in the UP and 8246.5 ± 2912.6 mg GAE/g of ExDW in the LP in wild jellyfish. Such values were not statistically significantly different from the phenolic content in the UP and LP from aquarium-grown jellyfish (1991.0 ± 390.2 and 5746.1 ± 125.5 mg GAE/g of ExDW, respectively) (Figure 3B). Phenolic compounds are synthesized and used in various physiological processes in higher plants, algae, and microalgae, including in the stress response, allowing the organism to interact with and adapt to its surrounding environment and to survive critical conditions such as UV radiation [77]. The higher content in phenolic compounds in reared jellyfish could be due to the possible stress to which these jellyfish are subjected by living under artificial conditions, including lighting. Since the hydroalcoholic extracts from reared and wild holobionts were quantitatively comparable (Section 2.2.1), the amount of phenolics per microalgae should be higher in reared holobionts than in wild holobionts, as wild holobionts contain a higher number of zooxanthellae per gram of DW (Table 2).

#### 2.3.3. Pigment Analysis in the Lipophilic Upper Phase (UP), Intermediate Phase (IP), and Hydrophilic Lower Phase (LP) of Hydroalcoholic Extracts

The three fractions of hydroalcoholic extract, lipophilic (UP), intermediate (IP), and hydrophilic lower phases, were analyzed by HPLC to identify and quantify the pigments of interest. Table 7 summarizes the concentration of chlorophyll-a and carotenoids detected in the UP, IP, and LP from wild and reared jellyfish, expressed as mg of pigment per gram of dry phase (UP-DW, IP-DW, and LP-DW).

Table 7 shows that no pigments were detected in the LP, as already reported in De Domenico et al. [15], definitively establishing that the LP is the aqueous phase in which proteins and other water-soluble components separate. Considering the pigment detected in the hydroalcoholic extract (Table 4) and in its fractions (Table 7), chlorophyll-a predominantly separates into the intermediate fraction (IP), while carotenoids predominantly separate into the lipophilic fraction (UP), except for b-carotene from the reared jellyfish samples, more of which was detected in the IP, even if in low quantities. In both wild and farmed holobionts, peridinin was found in the UP, where it was ten times higher in wild than in farmed specimens. Conversely, isoforms of peridinin were completely absent in the IP, both in wild and reared samples, which would suggest that peridinin–protein (PCP)-bound forms are present in the intermediate fraction, while free peridinin and probably its degradation forms may be separated into the lipophilic fraction.

As reported in De Domenico et al. [15], the lipophilic UPs were enriched in bioactive carotenoids. Here, we demonstrated that this protocol is also useful for the efficient extraction and separation of carotenoids from cultured jellyfish samples and for a possible use of these bioactive compounds. Indeed, jellyfish biomass could represent a good source of those compounds extensively used in applications such as nutraceuticals, cosmeceuticals, or biopharmaceuticals, and in the treatment and prevention of chronic and age-related diseases.

#### 2.3.4. Antioxidant Activity in Hydroalcoholic Extracts and in Its Lipophilic Upper Phase (UP) and Hydrophilic Lower Phase (LP)

The detection of carotenoids, phenolic compounds, and proteins suggests the presence of compounds with biological functionalities such as antioxidant activity. Therefore, the hydroalcoholic extracts and the derived fractions were analyzed for antioxidant activity by ABTS assay, which is considered more comprehensive as a preliminary test as it is applicable to both hydrophilic and hydrophobic (lipophilic) systems. Figure 4 shows the antioxidant activity measured in the 80% ethanol extracts and their upper phase (UP) and lower phase (LP) fractions expressed as nmol of TE (Trolox equivalents) per g of ExDW.

The antioxidant activity detected in the total hydroalcoholic extract (ExDW) was partially lost during fractionation, as UP and LP showed lower activity expressed per gram than ExDW, where LP was significantly greater than UP (Figure 4). Indeed, LPs from both populations had a higher antioxidant activity than the UPs, in line with the highest total phenol content (Figure 3B). The results also showed that the total hydroalcoholic extract (ExDW) of reared *C. andromeda* had a significantly higher antioxidant activity (15,588.2 ± 1447.7 nmol TE/g ExDW) than the antioxidant activity measured in the extract obtained from wild jellyfish (5824.4 ± 4051.1 nmol TE/g ExDW). The greater antioxidant activity in cultured jellyfish could be due to rearing conditions, which can induce different stresses. In particular, non-optimal simulated environmental conditions, such as lighting, could induce the holobiont to biosynthesize secondary metabolic compounds with antioxidant and/or anti-photo-oxidant properties. No significant differences in antioxidant activity between wild and reared jellyfish were detected in either UP or LP fractions. The total antioxidant activity data of wild jellyfish were comparable with data previously obtained on the oral arms and umbrellas of a different batch of wild *C. andromeda* specimens obtained with the same extraction method [15,24], considering that no separation of jellyfish body parts was performed here.

## 3. Materials and Methods

### 3.1. Rearing of Cassiopea Andromeda Jellyfish

*Cassiopea andromeda* rearing, including the conditions of polyp strobilation, the growth environment of ephyrae, and the maintenance of the reared jellyfish, was chosen on the basis of the information provided by the Genoa Aquarium, protocols in the literature, and previous experience in our laboratory. In a thermostatic room, ephyrae of *Cassiopea andromeda* (Forsskål, 1775) were obtained by the strobilation of polyps by gradually increasing the temperature of the seawater up to 29 °C. The polyps, kindly provided by the Genoa Aquarium, were kept in a small tank of 40 L of seawater. One week after strobilation induction, about a hundred ephyrae were liberated. Shortly after, they were transferred to a 200 L tank of seawater. Water conditions in the jellyfish tanks were kept at a constant temperature of 25 ± 1 °C and salinity at 36–39 ppt (parts per thousand). The light photoperiod was 12 h:12 h light/dark, and illumination was obtained using a FHO54W/Aquastar (Feilo Sylvania Italy S.p.A, Cinisello Balsamo, Milan, Italy), with a luminous flux of 2800 and a color temperature of 10,000 K, recreating the tropical light spectrum. Ephyrae and juvenile jellyfish were fed daily with fresh *Artemia salina* (Linnaeus, 1758) nauplii (2 g/100 L of seawater) harvested 24 h after hatching. Salinity was adjusted daily by the addition of deionized water to compensate for water loss by evaporation. Seawater change (50%) and basic cleaning of tank glass were undertaken once a week. Specimens of *C. andromeda* jellyfish were collected for experiments 3 months after strobilation. Two independent experiments were performed. After the biometric measurements, each individual was separately frozen in liquid nitrogen and stored at −80 °C.

### 3.2. Wild Specimen Collection

Live *C. andromeda* specimens were collected at the “Calamida” dock at “La Cala” marina, in the port of Palermo (Italy), using a hand net from September to December. For biochemical analysis, wild specimens measuring up to 9 cm in diameter were selected, in the size range of early summer ephyra generation, following the late spring peak of gonadal maturation [39,78]. The specimens were placed in a 50 L tank, and immediately after sampling (maximum 2 h), the jellyfish were brought to the laboratory to proceed with the biometric analysis. The gonads were dissected from the somatic tissue for each specimen, and each individual was separately frozen in liquid nitrogen and stored at −80 °C.

### 3.3. Symbiont Identification

Total DNA was extracted from 5 specimens collected in the Palermo harbor and from the laboratory population of *Cassiopea* polyps reared in the laboratory at the University of Salento, Lecce. About 20 mg of dry weight (DW) was used for DNA extraction with the GENEJEt Plant Genomic DNA purification kit following manufacturer instructions. After extraction, DNA was quantified with a microvolume UV-Vis Spectrophotometer NanoDrop™One/OneC (Thermo Scientific™, Waltham, MA, USA). For each specimen, we analyzed the chloroplast (cp) subunit rDNA sequence (cp23S-rDNA) of an approximately 0.7 kb region using the following primer pair: 23S1-M13 (5′-CACGACGTTGTAAAACGACGGCTGTAACTATAACGGTCC-3′) and 23S2-M13 (5′-GGATAACAATTTCACACAGGCCATCGTATTGAACCCAGC-3′). Reactions were carried out in a T-100 Thermocycler (Biorad, Hercules, CA, USA) under the following conditions: initial denaturing period of 4 min at 94 °C, 40 cycles at 94 °C for 1 min, 57 °C for 2 min, and 72 °C for 1 min, and a final extension period of 5 min at 72 °C. PCR products were purified by electrophoresis in 1.5% Tris–acetate (2 M Tris, 1 M acetic acid, 50 mM EDTA final concentration) agarose gel and visualized by SyBr DNA staining and long-wavelength ultraviolet light. PCR products were purified using Wizard^®^ SV Gel and PCR Clean-Up System (Promega, Madison, WI, USA) protocol according to the manufacturer’s directions. DNA was quantified and a PCR, followed by Sanger sequencing in an ABI Hitachi 3130 DNA Genetic Analyzer Sequencer (Applied Biosystems, Waltham, MA, USA), was conducted in a thermocycler under the following conditions: initial denaturing period of 1 min at 96 °C; 25 cycles at 96 °C for 10 s, 57 °C for 10 s, and 60 °C for 4 min. Samples were finally eluted through Sephadex columns prior to sequencing.

### 3.4. Symbiont Quantification

The quantification of symbionts in the jellyfish biomass was carried out as reported in De Domenico et al. [15] In brief, a known amount of the lyophilized powder of each *C. andromeda* jellyfish was put in a known volume of filtered seawater, vortexed, and stirred for 20 min. The determination of the concentration was carried out within 1 h using a Countess 3 Automated Cell Counter (Thermo Fisher Scientific Inc., Waltham, MA, USA), and autofluorescent cells were automatically detected, counted, and compared to the dry weight used. Representative images of the polyps and ephyra grown in the aquarium were captured using a Nikon SMZ25 stereomicroscope (Nikon Europe B.V., Amstelveen, The Netherlands) fitted with NIS-Elements imaging software (https://www.microscope.healthcare.nikon.com/en_EU/products/software/nis-elements accessed on 18 November 2024) and autofluorescence was investigated using Nikon intensilight C-HGFI fiber optic fluorescence (Nikon Europe B.V., Amstelveen, The Netherlands).

### 3.5. Hydroalcoholic Extraction and Its Fractionation into Phases

Lyophilized jellyfish samples were finely powdered with mortar and pestle in liquid nitrogen, and the dry powder was subjected to hydroalcoholic extraction and subsequent fractionation as reported in De Domenico et al. [15]. Briefly, jellyfish powders were put in 16 volumes (*w*/*v*) of 80% ethanol solution (80% EtOH) and placed on a rotary shaker (25 rpm) for 16 h, protected from light. Samples were then centrifuged (9000× *g* for 30 min at 4 °C); then, the supernatant were separated from the insoluble material and analyzed for pigments, lipids, fatty acids, protein, and phenolic compound content, as well as for antioxidant activity, as described below. The supernatant was then concentrated and lyophilized in light-protected conditions and at 4 °C in order to limit the loss of biological activity, as reported in De Domenico et al. [15], and subjected to the fraction separation protocol by cold-induced acetonitrile/water (ACN:H_2_O) phase separation to obtain a hydrophilic lower phase (LP) and a liposoluble upper phase (UP) separated by a semisolid intermediate phase (IP) as described in Leone [8].

The pellets remaining from the hydroalcoholic extraction, containing non-hydroalcoholic-soluble compounds, were hydrolyzed enzymatically, as described below.

### 3.6. Protein Hydrolysis

The insoluble residue after hydroalcoholic extraction was subjected to enzymatic hydrolysis as described in De Domenico et al. [20]. In brief, the insoluble fraction was subjected to an initial hydrolysis by pepsin from porcine gastric mucosa (P7012, Sigma-Aldrich, St. Louis, MI, USA) at a concentration of 1 mg/mL in 0.5 M acetic acid for 48 h at 4 °C and centrifuged (9000× *g* for 30 min) to isolate the supernatant containing pepsin-hydrolyzed (P) peptides. The resulting pellet was subjected to a second digestion with collagenase Type 1A from *Clostridium histolyticum* (Sigma-Aldrich, C9891) for 5 h at 37 °C, and after centrifugation (9000× *g* for 30 min), the supernatant contained the hydrolyzed collagen (C) peptides. In both pepsin and collagenase hydrolysis, the enzyme/substrate ratio was of 1:50 (*w*:*w*).

### 3.7. Protein Quantification and SDS-PAGE Analysis

The protein concentration was estimated in whole dry jellyfish biomass, in the hydroalcoholic extract (ExDW), in the UP and LP, and in the pepsin- and collagenase-hydrolyzed fractions by Bradford assay [79] using bovine serum albumin (BSA, Sigma-Aldrich 05470) as standard and modified as described by De Domenico et al. [15].

Total jellyfish proteins, ExDW, and hydrolyzed pepsin- and collagenase-hydrolyzed fractions were analyzed by SDS-PAGE using a FastCast premixed acrylamide solution 12% to prepare the gels and “All Blue Precision Plus Protein Standard” (Biorad, Hercules, CA, USA) as a molecular weight marker. In order to visualize protein bands, gels were both analyzed by a stain-free system with high sensitivity imaged using ChemiDoc™ MP Imaging System (Biorad, Hercules, CA, USA) and stained with Coomassie Brilliant Blue G-250 (Bio-Rad Protein Assay).

### 3.8. Pigment Identification and Quantification

The determination of pigment content was carried out by HPLC analysis for each sample, both on the hydroalcoholic extract (ExDW) and on the lipophilic (UP) and the hydrophilic (LP) fractions. The HPLC protocol, the instrumentation, and the relative calibration lines used are described in detail in De Domenico et al. [15].

### 3.9. Lipid Extraction and Fatty Acid Identification

Total lipids were extracted from 200 mg each of whole dry jellyfish biomass, hydroalcoholic extract (ExDW), lipophilic fraction UP, and from 20 mg of dried one-day-old *A. salina* nauplii using a modified Bligh and Dyer [80] method as described in De Rinaldis et al. [24]. The extracts were evaporated under nitrogen gas flow, esterified, and analyzed for lipid content and fatty acid composition by GC-MS analysis, as already reported [24].

### 3.10. Phenol Content and Antioxidant Activity Determination

The hydroalcoholic extract (ExDW) and the lipophilic (UP) and the hydrophilic (LP) fractions from each sample were analyzed by colorimetric methods for total phenol content and antioxidant activity using a modified Folin–Ciocalteu method [81] and the TEAC (Trolox Equivalent Antioxidant Capacity) method [82], respectively, as already reported in De Rinaldis et al. [24]. The assays were carried out in a 96-well microplate (Corning, Glendale, AZ, USA) by an Infinite M200 plate reader (Tecan Group Ltd., Männedorf, Germany) using appropriate blanks with the related solvents.

### 3.11. Statistical Analysis

Data were analyzed by the program Graph Pad Prism v5.0. Biometric data and other data to compare the two populations (wild vs. reared) were analyzed by an unpaired two-tailed *t*-test (*p* < 0.05). The number of microalgae per gram of dry and fresh biomass was analyzed by a two-way ANOVA test followed by Tukey’s multiple-comparison test (*p* < 0.05); pigment quantification in the WJF, ExDW, UP, and IP and data expressed per microalgae, as well as the composition of fatty acids, were analyzed by a two-way ANOVA test followed by a Bonferroni post-test (*p* < 0.05). Protein and total phenol content, as well as antioxidant activity data, were analyzed using a one-way ANOVA test followed by Tukey’s multiple comparison test (*p* < 0.05).

## 4. Conclusions

The richness of bioactive compounds in the holobiont *C. andromeda* was previously demonstrated [15,24]. Although it is a non-indigenous species, records of this jellyfish are being increasingly reported in the eastern to central Mediterranean Sea. Consistently with our previous findings, specimens of *C. andromeda* jellyfish sampled in the port of Palermo (Italy) exhibit a rich composition of protein, fatty acid, pigments, and phenolic compounds in their hydroalcoholic extracts and their lipophilic and hydrophilic fractions, as well as a notable antioxidant activity [15]. We also demonstrated a differential distribution of these bioactive compounds in the oral arms with respect to the umbrella, and ultimately the possibility of extracting quite easily the main photosynthetic pigments in the lipophilic phases and some valuable unsaturated fatty acids characteristic of the Symbiodiniaceae family or derived from the symbiosis with the gelatinous host.

To assess the potential of *C. andromeda* as a potential biofactory, here, we qualitatively and quantitatively evaluated differences in the production of bioactive compounds and their antioxidant activities between wild-harvested holobiont specimens compared to laboratory-reared specimens. We demonstrated that the number and concentration of zooxanthellae differ between wild and farmed holobionts, as do those of the pigments, fatty acids, and phenols produced. Also, heterotrophic nutrition influences the fatty acid composition, and probably both the density and metabolism, of zooxanthellae. The fatty acid composition of *Artemia salina* is variable, especially in the long-chain essential fatty acids component, suggesting that it is environmentally determined and also affected by the cyst storage conditions. The fatty acid profile is also a quality parameter of holobiont biomass; therefore, heterotrophic nutrition assumes a great importance in the possible breeding of jellyfish, both in terms of product quality and in economic terms; finding alternatives to *Artemia salina* nauplii is certainly another interesting field of research.

Despite the diversity found in the zooxanthellae strains present in the two populations analyzed here, this comparative analysis will help in the identification of the key culture parameters to be considered for producing the optimal breeding environment so as to promote a high growth rate of the holobiont and the greatest accumulation of bioactive substances produced both by the endosymbionts and the jellyfish host.

Recently, different free-living microalgal species (among which also *Symbiodinium* sp.) have received much attention due to their ability to produce novel bioactive metabolites, including carotenoids and nutritionally important fatty acids, that can provide health and cosmetic benefits [83,84,85]. Several environmental factors, such as temperature and nutrient concentration, strongly affect the type of fatty acids and pigments produced by autotrophic microalgae [86]. Tsirigoti et al. [87] observed a positive effect of low temperatures on biomass growth and total lipid accumulation, as well as an accumulation of the w3 DHA in nitrogen deprivation conditions in a cultivated *S. microadriaticum* strain and free-living *S. voratum*. Both triacylglycerol and cholesterol ester were enriched with PUFAs (DHA, EPA, and oleic acid) in free-living *Symbiodinium* spp. clade B upon nitrogen deprivation [88], and *Symbiodinium* clade C isolated from sea anemones showed a high concentration of peridinin and chlorophyll-a when grown at a temperature of 23 °C [89]; meanwhile, the dinoflagellate *S. voratum* showed a maximal biomass yield for the isolation of peridinin when was grown immobilized on a Twin-Layer PBR than when it was in a suspension culture [90]. Nonetheless, the cultivation of autotrophic dinoflagellates on a technical scale is restrained by a series of problems related to low growth rates and sensitivity to shear stress, and among microalgae, dinoflagellates have been further shown to be the most sensitive, even though marked differences between species still exist [91]. A possible solution would be to cultivate them together with their heterotrophic symbiont organisms under defined controlled conditions, such as the zooxanthellate *C. andromeda* jellyfish, whose biomass has already been demonstrated to be a potential source of bioactive substances, also deriving from the symbiosis with dinoflagellates. The diversity found in the strains and number of zooxanthellae present in the two analyzed populations deserve further study. The husbandry of *C. andromeda* and its endosymbiotic zooxanthellae could represent a new method of obtaining valuable compounds, whose value and amount could be modulated by a careful choice of environmental conditions, as well as the selection of the symbionts present in the holobionts. Moreover, both the hydroalcoholic extract and the two derived phases (lipophilic and hydrophilic) obtained by our protocol represent a mixture of concentrate molecules, such as essential fatty acids and carotenoids (mainly in the UP), and phenolic compounds (in the LP). These molecules have already been shown to have countless applications in the pharmaceutical, nutraceutical, and cosmeceutical fields, and their effects are currently being tested in vitro for their activity on cancer cell cultures (paper in preparation).

Our recent observations clearly demonstrate that *C. andromeda* is able to live and proliferate in specific highly trophic environments of the central Mediterranean. Based on data from the literature [92,93,94,95], an in-depth study of the most appropriate growth conditions in nature, including the presence of nutrients and of parameters which can influence the life cycle (strobilation conditions, growth), such as UV radiation, lighting, or temperature, will provide useful data for the design and optimization of confined rearing systems. Furthermore, additional laboratory studies on the influence of such parameters, or of stressors such as starvation, UV irradiation, etc., on the metabolism of the symbiotic association in *C. andromeda* will provide data for the optimization of rearing aimed at the production of primary metabolites (e.g., proteins, lipids, carbohydrates) or specific secondary metabolites with proven biological activities.

In this preliminary study, which to the best of our knowledge is the first in this field, some differences between wild and reared jellyfish are explored, indicating some key points in the enrichment of *C. andromeda* in bioactive compounds. An increase in the number of zooxanthellae per individual mass results in protein content enrichment, whereas the modulation of environmental parameters (lighting, temperature) could influence pigment and antioxidant content. Because of the unique characteristics of the epibenthic species *Cassiopeia andromeda*, its ease of rearing, its high reproductive success, and the quantity and variety of chemical species with biological activity and their potential uses, the breeding of *C. andromeda* is both desirable and potentially possible as an example of sustainable aquaculture, once the ecological issues associated with alien species control have been addressed.

## Figures and Tables

**Figure 1 marinedrugs-23-00019-f001:**
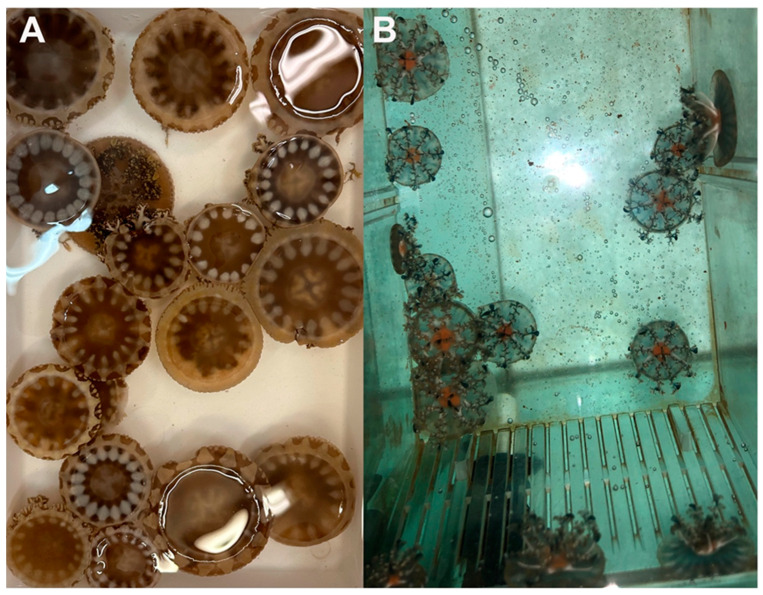
(**A**) *Cassiopea andromeda* collected from the port of Palermo (wild) and (**B**) *C. andromeda* born by strobilation and raised in the aquarium under controlled laboratory conditions (reared).

**Figure 2 marinedrugs-23-00019-f002:**
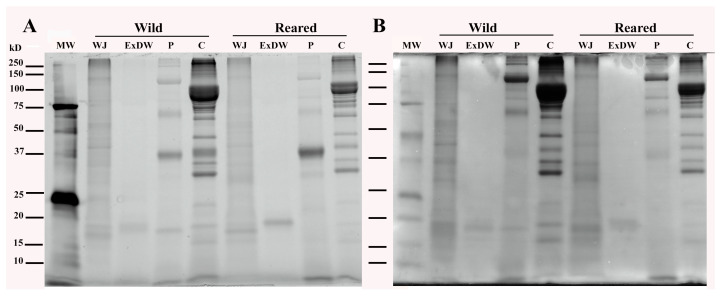
SDS-PAGE analysis of proteins from wild and reared *Cassiopea andromeda* holobionts imaged with ChemiDoc MP Imaging System (**A**) or stained with Coomassie Brilliant Blue (**B**). MW: molecular-weight size marker; WJ: whole-jellyfish proteins; ExDW: 80% EtOH soluble proteins; P: insoluble proteins hydrolyzed with pepsin; C: insoluble proteins hydrolyzed with collagenase. Each line was loaded with 20 mg of proteins.

**Figure 3 marinedrugs-23-00019-f003:**
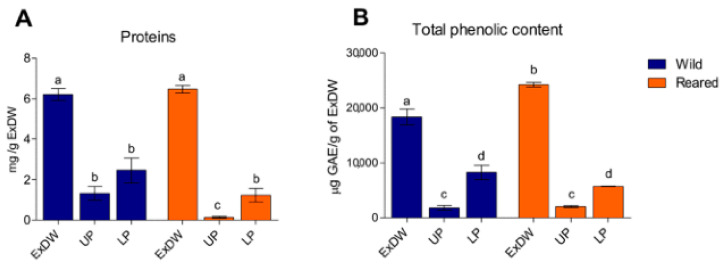
Protein (**A**) and total phenolic (**B**) contents evaluated in hydroalcoholic extracts (ExDW), upper phases (UP), and lower phases (LP) from wild and reared jellyfish, expressed as mg of proteins per g of ExDW and mg GAE/g of ExDW. Data between the two groups of samples were analyzed by unpaired *t*-test (two-tailed, *p* < 0.05). One-way ANOVA test (*p* < 0.05) was used to compare the data in all the phases (UPs and LPs). Different letters indicate significant differences.

**Figure 4 marinedrugs-23-00019-f004:**
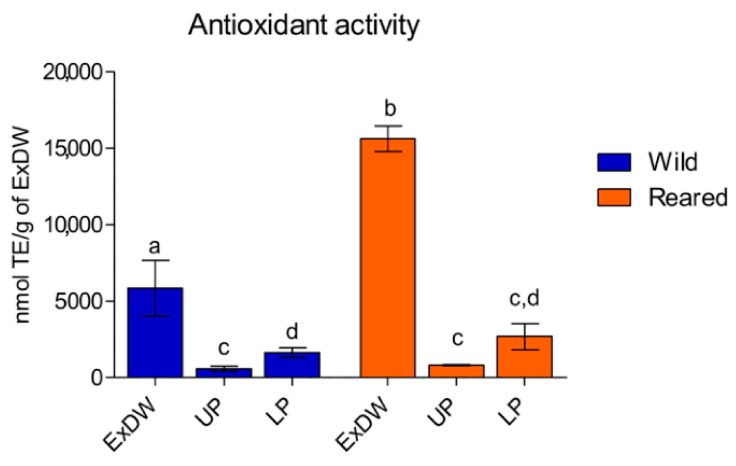
Antioxidant activity in total 80% ethanol extracts (ExDW) and in their fractions (upper phase, UP, and lower phase, LP) from wild and reared jellyfish, expressed as nmol TE per g of ExDW. Data between the two groups of samples were analyzed by an unpaired *t*-test (two-tailed *p* value); a one-way ANOVA test was used to compare the data in all the phases (UP and LP). Different letters indicate significant differences.

**Table 1 marinedrugs-23-00019-t001:** Biometric data of analyzed wild and reared specimens of *Cassiopea andromeda* jellyfish. DW: dry weight; JF: jellyfish; FW: fresh weight; OA: oral arm; UMB: umbrella. Data are means (n = 20) of two independent experiments and were analyzed by unpaired *t*-test with two-tailed *p* value (*p* < 0.05). Different superscript letters indicate significant differences.

Biometric Data of Wild and Reared *Cassiopea andromeda*							
Jellyfish Specimens	UMB Diameter (cm)	OA Diameter (cm)	Total FW (g)	FW/Diameter Ratio	Total DW (g)	DW/Diameter Ratio	DW (%FW)
Wild Mean ± SD	8.42 ± 0.73 ^a^	11.67 ± 1.66 ^c^	90.23 ± 31.44 ^e^	7.59 ± 2.10 ^g^	5.12 ± 1.82 ^i^	0.43 ± 0.13 ^m^	5.66 ± 0.24 ^o^
Reared Mean ± SD	6.42 ± 0.86 ^b^	9.00 ± 0.63 ^d^	17.72 ± 2.46 ^f^	1.96 ± 0.16 ^h^	0.79 ± 0.11 ^l^	0.09 ± 0.01 ^n^	4.50 ± 0.15 ^p^

**Table 2 marinedrugs-23-00019-t002:** Number of microalgae in wild or reared *Cassiopea andromeda* jellyfish expressed per gram of dry weight (microalgae/g WJ-DW) or fresh weight (microalgae/g WJ-FW). Specimen data were analyzed by two-way ANOVA test followed by Tukey’s multiple comparison test (*p* < 0.05). Asterisks indicate level of statistical significance. ** *p* ≤ 0.01 between all jellyfish samples. Data from the two populations were analyzed by unpaired *t*-test with two-tailed *p* value (*p* < 0.05). Superscript letters indicate significant differences among samples.

Number of Endosymbiont Microalgae			
	Jellyfish Samples	N. Microalgae/g WJ-DW	N. Microalgae/g WJ-FW
Wild	CAS_W1	55.7 × 10^6^	2.99 × 10^6^
	CAS_W2	96.2 × 10^6^ **	5.63 × 10^6^ **
	CAS_W3	65.6 × 10^6^	3.48 × 10^6^
	CAS_W4	60.1 × 10^6^	3.25 × 10^6^
	CAS_W5	25.7 × 10^6^ **	1.47 × 10^6^ **
	**Mean ± SD**	**60.6 ± 25.1 × 10^6 a^**	**3.36 ± 1.49 × 10^6 c^**
Reared	Cas_R1	26.7 × 10^6^	1.15 × 10^6^
	Cas_R2	25.2 × 10^6^	1.17 × 10^6^
	Cas_R3	19.8 × 10^6^	0.89 × 10^6^
	**Mean ± SD**	**23.9 ± 8.6 × 10^6 b^**	**1.07 ± 0.38 × 10^6 d^**

**Table 3 marinedrugs-23-00019-t003:** Protein amount in hydroalcoholic extracts (ExDW-soluble proteins) and in fractions obtained by pepsin or collagenase hydrolysis of insoluble materials from wild and reared *C. andromeda* jellyfish samples. Data are expressed as mg of protein per gram of whole jellyfish dry weight (WJ-DW). Data are means ± SD.

	Protein Content				
*C. andromeda* Holobiont Specimens	ExDW- Soluble Proteins	Pepsin- Hydrolyzed Proteins (P)	Collagenase-Hydrolyzed Proteins (C)	Undigested Residual Material	Total
	mg/g of WJ-DW				
Wild	5.64 ± 1.86	21.09 ± 19.02	7.55 ± 3.17	8.15 ± 0.66	42.44 ± 20.26
Reared	5.11 ± 0.08	7.61 ± 2.27	11.03 ± 2.04	2.45 ± 3.08	26.20 ± 7.80

**Table 4 marinedrugs-23-00019-t004:** Chlorophyll-a and carotenoid quantification in whole-jellyfish biomass (expressed as mg/g of WJ-DW), in hydroalcoholic extracts (expressed in mg/g of ExDW), and per number of microalgae (expressed in pg/microalgae) in wild (*n* = 5) and reared (*n* = 3) *C. andromeda* jellyfish biomass. Data are analyzed by two-way ANOVA test followed by Bonferroni post-test (*p* < 0.05). Different superscript letters indicate significant differences between pigments of wild and reared samples.

	Pigment Content in Whole Jellyfish Biomass			
*C. andromeda* Holobiont	Pigments	mg/g of WJ-DW	mg/g of ExDW	pg/Microalgae
Wild	*Chlorophyll a*	695.9 ± 358.1	1327.4 ± 483.1 ^a^	12.41 ± 6.35
	b-*carotene*	3.3 ± 1.2	6.6 ± 1.9	0.06 ± 0.04
	*Diadinoxanthin*	46.0 ± 4.5	82.2 ± 54.9	0.78 ± 0.60
	*Peridinin 1*	81.8 ± 4.3	136.2 ± 94.4	1.16 ± 0.70
	*Peridinin 2*	11.1 ± 8.8	19.6 ± 8.6	0.17 ± 0.06
	*Peridinin 3*	15.9 ± 2.4	26.2 ± 12.5	0.21 ± 0.10
	*Lutein*	419.7 ± 249.5	776.7 ± 128.3 ^c^	6.86 ± 2.67 ^e^
Reared	*Chlorophyll a*	237.3 ± 25.2	598.9 ± 128.2 ^b^	10.19 ± 2.76
	b-*carotene*	1.9 ± 0.4	4.8 ± 0.6	0.08 ± 0.01
	*Diadinoxanthin*	27.1 ± 1.8	68.0 ± 10.5	1.15 ± 0.17
	*Peridinin 1*	34.0 ± 1.1	86.1 ± 19.4	1.44 ± 0.25
	*Peridinin 2*	8.3 ± 0.5	20.8 ± 3.3	0.35 ± 0.08
	*Peridinin 3*	7.3 ± 0.4	18.6 ± 4.4	0.31 ± 0.06
	*Lutein*	61.9 ± 3.8	167.1 ± 47.5 ^d^	2.77 ± 0.30 ^f^

**Table 5 marinedrugs-23-00019-t005:** Total lipids detected in whole jellyfish (WJ), hydroalcoholic extract (ExDW), and its lipophilic upper phase (UP) and hydrophilic lower phase (LP) obtained by fractionation of hydroalcoholic extracts from reared *Cassiopea andromeda* jellyfish. Data from wild *C. andromeda* are from [15,24] ^1^. Data are means of 3 independent experiments. Data were analyzed by unpaired *t*-test with two-tailed *p* value (*p* < 0.05). ND, not detected. Different superscript letters indicate significant differences.

	Total Lipid Contents			
*C. andromeda* Holobiont Specimens	Whole Jellyfish (WJ)	Total Hydroalcoholic Extracts (ExDW)	Lipophilic Fraction (UP) of ExDW	Hydrophilic Fraction (LP) of ExDW
	mg/g			
Wild	9.4 ± 0.4 mg/g of WJ-DW ^1 a^	15.5 ± 0.5 mg/g of ExDW ^1 c^	10.2 ± 0.7 mg/g of ExDW ^1 c^	ND ^1^
Reared	605.8 ± 305.0 mg/g of WJ-DW ^b^	549.3 ± 225.9 mg/g of ExDW ^d^	3442.3 ± 1160.6 mg/g of ExDW ^e^	ND

^1^ De Rinaldis et al. [24]; De Domenico et al. [15].

**Table 6 marinedrugs-23-00019-t006:** Fatty acid (FA) composition of whole reared *Cassiopea andromeda* jellyfish (WJ), 80% ethanol extract (ExDW), and its lipophilic (UP) and hydrophilic (LP) phases obtained by fractionation of hydroalcoholic extracts and *Artemia salina* nauplii, used to feed reared jellyfish. Data are means of 3 independent experiments and are expressed as percentages of total fatty acids. Data were analyzed by two-way ANOVA test, followed by Bonferroni post-test (*p* < 0.05). Asterisks indicate level of statistical significance between whole jellyfish (WJ) and other samples (* *p* ≤ 0.05, ** *p* ≤ 0.01, and *** *p* ≤ 0.001).

Fatty Acid Composition in Reared *C. andromeda* Holobionts and in *A. salina*					
Fatty Acid (FA)	Whole Jellyfish (WJ)	80% EtOH Extract (ExDW)	Upper Phase (UP)	Lower Phase (LP)	*Artemia salina*
Saturated FA (SFA) %					
Lauric acid *C12:0*	-	-	-	-	-
Myristic acid *C14:0*	2.7 ± 0.7	3.6 ± 0.6	-	-	3.9 ± 1.2
Pentadecanoic acid *C15:0*	-	-	-	-	0.5 *** ± 1.0
Palmitic acid *C16:0*	30.0 ± 2.7	23.6 ± 1.6	30.3 ± 1.6	-	18.1 *** ± 0.9
Margaric acid *C17:0*	0.7 ± 2.2	0.7 ± 0.2	-	-	1.0 ± 0.3
Stearic acid *C18:0*	31.1 ± 7.1	22.0 ± 4.6 ***	27.2 ± 2.8	-	6.9 *** ± 1.5
Arachidic acid *C20:0*	2.2 ± 0.6	1.3 ± 0.1	3.0 ± 0.8	-	0.1 ± 0.0
**Total SFA**	**66.7 ± 9.2**	**51.2 ± 3.4 *****	**60.5 ± 2.9 ****	**-**	**33.6 ± 0.2**
Monounsaturated FA (MUFA) %					
Palmitoleic acid *C16:1* (ω7)	1.5 ± 0.4	2.1 ± 0.3	-	-	11.8 *** ± 1.8
Margaroleic acid *C17:1*	-	-	-	-	3.0 *** ± 0.1
Oleic acid *C18:1 cis-9*	-	-	-	-	25.7 *** ± 2.5
Isoleic acid *C18:1 trans-10*	1.5 ± 0.9	2.1 ± 0.1	2.0 ± 0.5	-	-
Vaccenic acid *C18:1 trans-11*	2.4 ± 0.2	-	-	-	0.3 *** ± 0.0
Gondonic acid *C20:1 cis-11*	-	-	-	-	0.4 *** ± 0.0
Paullic acid *C20:1 cis-13*	-	-	-	-	0.5 *** ± 0.0
**Total MUFA**	**5.4 ± 1.0**	**4.2 ± 0.4**	**2.0 ± 0.5**	**-**	**41.7 *** ± 0.0**
Polyunsaturated FA (PUFA) %					
Linoleic acid *C18:2 cis-9,12* (ω6)	1.2 ± 0.3	1.5 ± 0.2	-	-	5.1 ± 0.1
Isolinoleic acid *C18:2 cis-6,9* (ω9)	-	-	1.0 ± 0.0	-	-
a-Linolenic acid *C18:3 cis-9,12,15* (ω3)	0.9 ± 0.3	0.9 ± 0.2	-	-	-
g-Linolenic acid *C18:3 cis-6,9,12* (ω6)	2.6 ± 0.7	3.8 ± 0.5	3.1 ± 0.5	-	11.8 *** ± 0.2
Stearidonic acid *C18:4* (ω3)	7.2 ± 2.2	14.8 ± 2.3 *	15.7 ± 1.9 *	-	1.7 ± 0.0
Eicosadienoic acid *C20:2* (ω6)	-	-	-	-	-
Dihomo-γ-linolenic acid *C20:3* (ω6)					0.1 *** ± 0.6
Arachidonic acid *C20:4* (ω6)	9.1 ± 1.7	13.5 ± 0.3 *	11.3 ± 1.4 *	-	1.4 *** ± 0.3
Eicosapentaenoic acid *C20:5* (ω3)	2.0 ± 1.4	2.8 ± 0.9	3.1 ± 0.3 *	-	7.5 ± 0.3
Docosatetraenoic acid *C22:4* (ω6)	1.5 ± 0.2	1.0 ± 0.3	-	-	-
Docosapentaenoic acid *C22:5* (ω6)	1.4 ± 0.2	-	-	-	-
Docosahexaenoic acid *C22:6* (ω3)	2.0 ± 0.6	6.3 ± 0.9 *	3.3 ± 0.3 *	-	0.2 ± 0.8
**Total PUFA**	**27.9 ± 8.4**	**44.6 ± 2.6 *****	**37.5 ± 3.1 *****	**-**	**27.7 ± 0.1**
**Total fatty acids (%)**	**100.0**	**100.0**	**100.0**	**0**	**100.0**
*Σω6*	16.3	19.7	15.8		18.4
*Σω3*	12.2	25.0	22.2		9.3
Ratio *ω*6/*ω*3	*1.3*	*0.8*	*0.7*		*2.0*

**Table 7 marinedrugs-23-00019-t007:** Chlorophyll-a and carotenoid quantification in the lipophilic UP, IP, and hydrophilic LP phases obtained by hydroalcoholic extraction and fractionation from wild and reared specimens of *Cassiopea andromeda.* Data are expressed as means of μg of pigment per gram of the relative phase (dry weight). Data were analyzed by two-way ANOVA test, followed by Bonferroni post-test (*p* < 0.05). Different superscript letters indicate significant differences between pigments of wild and reared samples.

	Pigment Content in Fractions of Hydroalcoholic Extracts of *C. andromeda* Samples			
*C. andromeda* Holobionts		Lipophilic Upper Phase (UP)	Intermediate Phase (IP)	Hydrophilic Lower Phase (LP)
		mg/g of UP-DW	mg/g of IP-DW	mg/g of LP-DW
Wild	*Chlorophyll a*	7237.5 ± 5270.5	15,194.6 ± 14,064.6	-
	b-*carotene*	n.d.	426.9 ± 400.0	-
	*Diadinoxanthin*	647.2 ± 389.2	365.5 ± 288.3	-
	*Peridinin 1*	3369.6 ± 2511.6	516.0 ± 399.0	-
	*Peridinin 2*	483.2 ± 401.4	-	-
	*Peridinin 3*	665.1 ± 513.2	-	-
	*Lutein*	19,186.6 ± 159,392.6 ^a^	6276.4 ± 5998.1	-
Reared	*Chlorophyll a*	1506.8 ± 728.9	11,647.1 ± 4371.7	-
	b-*carotene*	8.4 ± 5.1	119.1 ± 73.3	-
	*Diadinoxanthin*	377.5 ± 124.8	109.2 ± 38.9	-
	*Peridinin 1*	315.4 ± 188.6	182.5 ± 96.8	-
	*Peridinin 2*	102.7 ± 27.6	-	-
	*Peridinin 3*	83.6 ± 20.6	-	-
	*Lutein*	535.2 ± 329.9 ^b^	n.d.	-

## Data Availability

Relevant data are contained within the article and Appendix A; raw data, photos, and videos are available from the corresponding author, A.L. Video Abstract. Video recording made with underwater ROV FIFISH V6 (QYSEA, Shenzhen, Guangdong, China) with 100 m of cable reel, remote controller and VR viewer. 4K video camera, 97 Wh battery, on-board telemetry instruments, autopilot and omnidirectional movements; control via App. Credits: Andrea Toso.

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
