# Peer review of "Wild or Reared? Cassiopea andromeda Jellyfish as a Potential Biofactory"

_marinedrugs, 2025, doi:10.3390/md23010019_

Round 1
Reviewer 1 Report
Comments and Suggestions for Authors
The article studied the potential of Cassiopea andromeda as a source of bioactive compounds. By comparing the wild and reared Cassiopea andromeda, it analyzed the contents of their biochemical composition, photosynthetic pigments, fatty acids, proteins, etc., as well as their antioxidant activities. It was found that there were significant differences between the two in multiple aspects, and the reared jellyfish had certain advantages in some respects, providing a basis for the sustainable cultivation of this species. The experimental design was in line with the experimental conclusions. I think the experiment was relatively complete, but some theoretical explanations need to be supplemented. I suggest that the paper still needs some revisions before it can be accepted.
1. Check the article for any spelling or grammar errors and ensure that all figures and tables are clear, accurate, and have appropriate titles and captions, such as "zooxanthellatae" (line 18), "hydroalcohoic" (line 233).
2. In lines 39 - 43, the potential applications of jellyfish in multiple fields are mentioned in the text, but the specific mechanisms of these applications are not discussed. It is recommended to supplement relevant background information to enhance the depth of the paper.
3. In lines 47 - 55, it is recommended to supplement more background information on the interaction between Cassiopea andromeda and symbiotic algae and how this symbiotic relationship affects the production of active compounds.
4. In the results of biomass characterization on line 164, the paper described in detail the differences in biomass, the number of symbiotic algae, biochemical composition, etc. between the wild and reared Cassiopea andromeda. When discussing these differences, although possible reasons such as the influence of environmental factors and nutritional modes were mentioned, the more in-depth discussion of the mechanisms was lacking. It is recommended to provide more discussion on the possible biological reasons behind these differences.
5. In the part of identification and quantification of microalgae pigments in the hydroalcoholic extract (line 310), it is recommended to supplement the reasons for the differences in pigment contents between the reared and wild jellyfish under different environmental conditions, so as to have more comprehensive understanding of the physiological and ecological characteristics of jellyfish and symbiotic algae under different environments and provide more valuable references for jellyfish cultivation and related research.
6. In the aspect of antioxidant activities (line 536), the article only compares the reared and wild jellyfish. It is recommended to supplement the comparison with other antioxidant active substances to prove its value.
7. In lines 563 - 579, please supplement the background information on choosing these conditions (temperature, light, etc.) as the growth environment of reared jellyfish and evaluate the influence of these factors on the growth of jellyfish and the production of bioactive compounds.
8. Please supplement the specific brands of the reagents used in the experiment to increase the repeatability of the experiment, such as pepsin (line 641), collagenase (line 643), BSA (line 650).
9. In the conclusion on lines 687 - 692, the potential of jellyfish as a source of bioactive compounds was summarized. It is recommended to further discuss the potential impact of these findings on future cultivation practices and the extraction of bioactive compounds.
10. In the discussion part, regarding "the influence of heterotrophic nutrition on fatty acid composition" (line 704), the influence of feeding nauplii of copepods was mentioned in the text, but its specific mechanism was not introduced in detail. It is recommended to supplement relevant background information.
11. It is recommended that the authors further discuss how their findings are related to the existing literature and the contribution of these findings to the fields of jellyfish cultivation and bioactive compound research in the discussion part.
12. In the conclusion part, the results of this study can be compared and analyzed more in-depth with those of other similar jellyfish or marine organisms to highlight the unique contribution of this study and its position in the entire field. Meanwhile, it is also possible to discuss how to integrate the results of this study with the research in other related fields (such as marine ecological protection, sustainable development of aquaculture, etc.) to achieve broader scientific value and social and economic benefits.
Author Response
Reviewer 1
The Authors thank the Reviewer for the time and competencies dedicated to review the manuscript and for useful comments and suggestions.
Please find the detailed responses below and the corresponding revisions and/or corrections highlighted/in track changes in the re-submitted files.
Comments 1. Check the article for any spelling or grammar errors and ensure that all figures and tables are clear, accurate, and have appropriate titles and captions, such as"zooxanthellatae" (line 18), "hydroalcohoic" (line 233).
Response 1: Thanks for your careful reading, we have corrected the errors you reported and others that had escaped our last reading.
Comment 2. In lines 39 - 43, the potential applications of jellyfish in multiple fields are mentioned in the text, but the specific mechanisms of these applications are not discussed. It is recommended to supplement relevant background information to enhance the depth of the paper.
Response 2: We thank for pointing out this oversight. A paragraph was added summarizing the main field of applications of the main bioactive compounds
Comment 3. In lines 47 - 55, it is recommended to supplement more background information on the interaction between Cassiopea andromeda and symbiotic algae and how this symbiotic relationship affects the production of active compounds.
Response 3: We fully agree with the reviewer and have modified the text. Text was added in the manuscript (in red).
Comment 4. In the results of biomass characterization on line 164, the paper described in detail the differences in biomass, the number of symbiotic algae, biochemical composition, etc. between the wild and reared Cassiopea andromeda. When discussing these differences, although possible reasons such as the influence of environmental factors and nutritional modes were mentioned, the more in-depth discussion of the mechanisms was lacking. Itis recommended to provide more discussion on the possible biological reasons behind these differences.
Response 4: Thank you for pointing this out. We agree with this comment. Therefore, we have added the following text (in red) to the lines 310 – 318:
“Since collagen, a structural protein of the host jellyfish, does not appear to vary much in the two holobiont populations (Table 3), it is possible to infer that the greater variability is due to proteins originating from the symbionts and/or from the symbiont-host metabolic interactions in the two different environmental conditions. Indeed, the difference in the number of symbionts between wild and farmed samples overlaps the difference in the content of non-collagenous fibrillar proteins, i.e., pepsin hydrolysed proteins, which in reared holobionts are about a third of those in wild specimens. Therefore, the number of zooxanthellae per dry weight could become a parameter to be considered to predict and/or increase the total protein content of the holobiont.”
Comment 5. In the part of identification and quantification of microalgae pigments in the hydroalcoholic extract (line 310), it is recommended to supplement the reasons for the differences in pigment contents between the reared and wild jellyfish under different environmental conditions, so as to have more comprehensive understanding of the physiological and ecological characteristics of jellyfish and symbiotic algae under different environments and provide more valuable references for jellyfish cultivation and related research.
Response 5: We fully agree with the referee, indeed we discussed the results based on our deductions and on the literature, we argued and interpreted the differences found in the qualitative-quantitative composition of pigments in wild and cultivated jellyfish populations. However, in our opinion, we cannot go further into the arguments on differences in the metabolism of holobionts because more specific experiments would be needed that are not included in this work. In agreement the text in the line 410-412 has been slightly changed.
Comment 6. In the aspect of antioxidant activities (line 536), the article only compares the reared and wild jellyfish. It is recommended to supplement the comparison with other antioxidant active substances to prove its value.
Response 6: I'm not sure I understood the reviewer's request, if a comparison with available data on jellyfish or if he means to have a generic antioxidant activity parameter, external for example to jellyfish, such as the antioxidant activity of ascorbic acid or other hydrophilic or lipophilic known antioxidants.
In the first case, in agreement with the reviewer comment we added the following text (in red at lines 615-618:
“The total antioxidant activity data of wild jellyfish were comparable with data previously obtained on oral arms and umbrellas of a different batch of wild C. andromeda specimens obtained with the same extraction method (15, 24), considering that here no separation of the jellyfish body parts was performed.”
In the second case, in general we avoid this kind of comparison due to the extreme variability of the antioxidant values available. This variability is due not only to the different sources and the lipophilic or hydrophilic nature of substances but also to the related extraction and measurement methods. Indeed, there is a variation in the antioxidant activity results due to the choice of solvents, extraction time, temperature, etc., and the antioxidant capacity of extracts is often measured with different assays and given in different units. Therefore, it may be inappropriate to compare the antioxidant activity with pure compounds or other extracts. One could say, for example, that the antioxidant activity in farmed jellyfish is 13 times that of ascorbic acid (vitamin C), or double that of the same amount of pumpkin seed extract, or 3 times that of apple juice, but we would consider this misleading. (Apak et al. Microchim Acta (2008) 160: 413–419, DOI 10.1007/s00604-007-0777-0; Fruhwirth et al. 2005 Anal Bioanal Chem, DOI 10.1007/s00216-005-0179-2)
Comment 7. In lines 563 - 579, please supplement the background information on choosing these conditions (temperature, light, etc.) as the growth environment of reared jellyfish and evaluate the influence of these factors on the growth of jellyfish and the production of bioactive compounds.
Response 7: Thanks for highlight this gap, the growth environment of the cultured jellyfish was chosen based on information from the Aquarium of Genoa, literature and previous experience in our laboratory, this information was added to the text (in red).
However, since different sets of parameters were not compared in this work, we did not consider or discuss the effects of different conditions (e.g., temperature, light, UV, etc.) on the growth of jellyfish and the production of bioactive compounds. There is other works that alredy did this studies (e.g., Kühnhold et al. Front. Mar. Sci., 10: 2023, https://doi.org/10.3389/fmars.2023.1048346; doi: 10.3389/fmars.2024.1348864, etc). Based on the results of the present work, this could be a good suggestion for future developments of this kind of studies.
Comment 8. Please supplement the specific brands of the reagents used in the experiment to increase the repeatability of the experiment, such as pepsin (line 641), collagenase (line 643), BSA (line 650).
Response 8: Thanks. Done. Text in red.
Comment 9. In the conclusion on lines 687 - 692, the potential of jellyfish as a source of bioactive compounds was summarized. It is recommended to further discuss the potential impact of these findings on future cultivation practices and the extraction of bioactive compounds.
Response 9: Thanks. Done. Text in red.
Comment 10. In the discussion part, regarding "the influence of heterotrophic nutrition on fatty acid composition" (line 704), the influence of feeding nauplii of copepods was mentioned in the text, but its specific mechanism was not introduced in detail. It is recommended to supplement relevant background information.
Response 10: Further background information was added also based on basic literature such as Bengtson, David & Léger, Philippe & Sorgeloos, Patrick. (1991). Use of Artemia as a food source for aquaculture. Artemia Biology. 29.
Comment 11. It is recommended that the authors further discuss how their findings are related to the existing literature and the contribution of these findings to the fields of jellyfish cultivation and bioactive compound research in the discussion part.
Response 11: Discussion of the results in relation to the appropriate literature has been done and as far as possible also updated in the conclusions section. We have already published the results on wild C. andromeda, where some data have been compared with literature data. However, since there are no works similar to the present work comparing wild and farmed zooxanthellate jellyfish, in the present manuscript, some specific data have been mostly exposed without having a comparison in the literature yet.
Comment 12. In the conclusion part, the results of this study can be compared and analyzed more in-depth with those of other similar jellyfish or marine organisms to highlight the unique contribution of this study and its position in the entire field. Meanwhile, it is also possible to discuss how to integrate the results of this study with the research in other related fields (such as marine ecological protection, sustainable development of aquaculture, etc.) to achieve broader scientific value and social and economic benefits.
Response 12: Thanks. Done.

Reviewer 2 Report
Comments and Suggestions for Authors
Review for the paper “Wild or reared? Cassiopea andromeda jellyfish as potential biofactory” by Stefania De Domenico, Andrea Toso, Gianluca De Rinaldis, Marta Mammone, Lara M. Fumarola, Stefano Piraino, Antonella Leone submitted to “Marine Drugs”.
The authors of this research paper conducted an analysis of the zooxanthellate jellyfish Cassiopea Andromeda from the Mediterranean Sea. They investigated this organism to determine its potential as a source of bioactive compounds, focusing on specimens collected from the wild as well as those reared under controlled laboratory conditions. By employing a standardized extraction protocol, they analyzed the biochemical composition of both the wild and cultured populations, examining parameters such as protein, lipid, and pigment contents, as well as the relative concentration of their dinoflagellate symbionts. The authors found that while both populations exhibited similar extraction yields, there were notable differences in the overall biomass, the quantity of zooxanthellae, and specific biochemical components like protein and lipid levels. Additionally, the fatty acid composition varied significantly between the two groups. The study revealed that the hydroalcoholic extracts obtained from jellyfish cultured in controlled settings demonstrated enhanced antioxidant activity. This increased effectiveness was attributed to a higher concentration of bioactive compounds in the cultivated jellyfish compared to their wild counterparts. The results of this study may have important implications for various industries, particularly those focused on food, nutraceutical, and pharmaceutical applications.
The paper is well-written and only minor revisions are required mostly to improve the discussion.
Recommendations.
Introduction. The authors should mention the key biological activities of the molecules identified from C. Andromeda. How do they compare to similar compounds found in other jellyfish species?
Introduction. The authors should mention the specific biotechnological applications for which the compounds derived from C. andromeda may be utilized.
Introduction. The authors should mention about the ecological implications of cultivating C. andromeda in areas where it is considered a non-native species.
Introduction. It would be useful to mention how the mutualistic relationship between C. andromeda and its symbiotic dinoflagellates potentially influence the biochemical profiles of the jellyfish.
Introduction. L 91. The authors should provide a clearer explanation of how temperature and light interact to influence the biochemical processes in both zooxanthellae and their jellyfish hosts.
Results and Discussion. Section 2.1.1. The authors should explain what specific environmental factors in the wild contributed to the larger sizes and higher biomass of C. andromeda compared to laboratory-reared specimens. They should discuss what the lower DW/FW ratio in reared jellyfish might indicate about their health or growth conditions could add depth to the analysis.
Results and Discussion. Section 2.1.2. The authors should discuss the significance of clade diversity among the zooxanthellae hosted by C. andromeda. In the context of the jellyfish's adaptive capabilities or resilience to environmental changes.
Results and Discussion. Section 2.2. The authors should explain the biological significance of the differences in soluble and insoluble protein content between wild and reared specimens.
Results and Discussion. The authors should mention the potential applications of the biologically active compounds extracted from C. andromeda.
Results and Discussion. The authors should explain what factors might contribute to the observed qualitative differences in protein profiles between wild and reared jellyfish.
Results and Discussion. The authors should discuss the implications of finding differing concentrations of chlorophyll-a and lutein between wild and reared C. andromeda. In particular, how these differences might affect the ecological roles of these jellyfish in their respective environments, such as their energy dynamics and trophic interactions.
Results and Discussion. What specific environmental factors in the natural habitat of C. andromeda might contribute to the observed variability in pigment concentrations?
Results and Discussion. Section 2.3. The authors should discuss how the feeding practices, such as using Artemia salina, might impact not only the fatty acid composition but also the overall health and survival of reared C. andromeda
Results and Discussion. The authors should discuss the variations in fatty acid profiles between the two jellyfish populations in terms of stress responses under different environmental conditions.
Results and Discussion. What potential mechanisms could explain the observed enrichment of polyunsaturated fatty acids (PUFAs) during hydroalcoholic extraction, particularly in essential fatty acids such as arachidonic acid and docosahexaenoic acid?
Results and Discussion. Given that the protein content is similar across the two populations, the authors should discuss the implications of the different protein distributions between the UP and LP phases in reared jellyfish compared to wild jellyfish.
Results and Discussion. Given the effect of artificial lighting on phenolic content, the authors should offer specific lighting adjustments to reduce stress and optimize the growth of reared C. andromeda
Results and Discussion. The authors should explain more clearly what factors might contribute to the significant difference in peridinin concentrations between wild and farmed C. andromeda specimens.
Results and Discussion. Given the higher antioxidant activity in the total hydroalcoholic extract of reared jellyfish, the authors should propose some mechanisms that might underlie this observed increase.
Methods. Section 3.2. The authors should indicate the sampling period for wild specimens.
Methods. Section 3.11. Did the authors check the data for normality and homogeneity of variance prior to ANOVA?
Author Response
Reviewer 2
The authors thank the reviewer for the time dedicated to the manuscript, for the appreciation of the work done and for the useful comments on the manuscript, important for its improvement.
Below we answer your recommendation point by point.
Comment 1: Introduction. The authors should mention the key biological activities of the molecules identified from C. andromeda. How do they compare to similar compounds found in other jellyfish species?
Response 1: The key compounds identified in the holobiont C. andromeda are typical of zooxanthellate jellyfish, namely fatty acids, pigments, carotenoids, phenolic compounds and proteins such as collagen. Each of that compound have a plethora of biological activities both in the organisms in which they are synthetized/metabolized and in humans as nutraceuticas, pharmaceuticals as extracted compounds. To focus on the main objective of the paper, which is the comparison between wild and reared population of C. andromeda, we introduced in lines 48-54 the main class of potentially bioactive compounds, in lines 78-86 and 103-113 we focused on specific compounds coming from the hosted zooxantellae.
In our opinion, to introduce all the specific biological activities, as well as making a simple list would unnecessarily lengthen the introduction. In addition, we are writing another article on the biological activity of hydroalcoholic extracts of C. andromenda and C. tuberculata, the introduction of which will focus on that topic.
Comment 2: Introduction. The authors should mention the specific biotechnological applications for which the compounds derived from C. andromeda may be utilized.
Response 2: Thanks. Done. Text (in red) added in the manuscript.
Comment 3: Introduction. The authors should mention about the ecological implications of cultivating C. andromeda in areas where it is considered a non-native species.
Response 3: Agree. Text (in red) added in the manuscript.
Comment 4: Introduction. It would be useful to mention how the mutualistic relationship between C. andromeda and its symbiotic dinoflagellates potentially influence the biochemical profiles of the jellyfish.
Response 4: : We fully agree with the reviewer. A paragraph was added at lines 80-86.
Comment 5: Introduction. L 91. The authors should provide a clearer explanation of how temperature and light interact to influence the biochemical processes in both zooxanthellae and their jellyfish hosts.
Response 5: We agree in principle. However, this paper focuses on a preliminary comparison of the main biochemical composition of the two jellyfish populations. Therefore, a more specific introduction on the factors that influence the biochemical processes in both zooxanthellae and jellyfish, in our opinion, may be inappropriate and/or too premature.
Comment 6: Results and Discussion. Section 2.1.1. The authors should explain what specific environmental factors in the wild contributed to the larger sizes and higher biomass of C. andromeda compared to laboratory-reared specimens. They should discuss what the lower DW/FW ratio in reared jellyfish might indicate about their health or growth conditions could add depth to the analysis.
Response 6: We agree in principle, however there is so little information in the literature on natural growth of C. andromeda in the Mediterranean, and to our knowledge, our study is the first to make such a comparison between wild and reared jellyfish, so it is difficult to speculate without other specific studies. We have indicated this in the manuscript (text in red, lines 215-217).
“These differences are certainly due to different environmental conditions, including the availability of different nutrients, although specific experiments are still needed to focus on the specific parameters responsible.”
Comment 7: Results and Discussion. Section 2.1.2. The authors should discuss the significance of clade diversity among the zooxanthellae hosted by C. andromeda. In the context of the jellyfish's adaptive capabilities or resilience to environmental changes.
Response 7: Thanks for the suggestion, this could be a good experimental hypothesis in the next steps. In our opinion, extending the discussion beyond the real meaning of our results is not appropriate at this stage.
Comment 8: Results and Discussion. Section 2.2. The authors should explain the biological significance of the differences in soluble and insoluble protein content between wild and reared specimens.
Response 8: We fully agree with the reviewer. In paragraph 2.2.2 some text has been added, in red.
Comment 9: Results and Discussion. The authors should mention the potential applications of the biologically active compounds extracted from C. andromeda.
Response 9: We thank for pointing out this. The potential applications of the biologic active compounds were mentioned in the introduction following the suggestion of the Reviewer 1.
Comment 10: Results and Discussion. The authors should explain what factors might contribute to the observed qualitative differences in protein profiles between wild and reared jellyfish.
Response 10: The analysis of the electrophoretic separation of total proteins, is mainly qualitative and gives only a comparative value but does not allow to distinguish and identify the proteins/polypeptides nor their origin (host or symbiont). The only discussion that we could have done on the basis of the molecular weights of the bands and the known proteins of the symbionts, was made mentioning LCH. Further discussions would be highly speculative if based only on SDS-PAGE analysis.
Comment 11: Results and Discussion. The authors should discuss the implications of finding differing concentrations of chlorophyll-a and lutein between wild and reared C. andromeda. In particular, how these differences might affect the ecological roles of these jellyfish in their respective environments, such as their energy dynamics and trophic interactions.
Response 11: We believe that further experiments are needed to measure and compare trophic dynamics in the holobiont of C. andromenda. We deliberately wanted to emphasize only the comparison and postpone a discussion on the type of pigments, the trophic flows and its significance to more specific studies already planned in our laboratory.
Comment 12: Results and Discussion. What specific environmental factors in the natural habitat of C. andromeda might contribute to the observed variability in pigment concentrations?
Response 12: In the text we mentioned that the low lutein content due to the stress of the non-natural environment, such as the applied artificial light and the concentration of nitrogen which influence its biosynthesis. Also, some environmental factors such as lighting, temperature, nutrients, etc. can influence the number and the metabolism of different photosynthetic microorganisms. Indeed, we considered that the quantity of chlorophyll-a, specific to Symbiodinium sp., is the same in the two populations when expressed in picograms of pigment/microalgae. Therefore, the lower amount of chlorophyll-a in reared jellyfish as compared to the wilds is therefore due to a lower presence of Symbiodiniaceae in their biomasses.
Comment 13: Results and Discussion. Section 2.3. The authors should discuss how the feeding practices, such as using Artemia salina, might impact not only the fatty acid composition but also the overall health and survival of reared C. andromeda
Response 13: We believe we do not have data to discuss in these terms. In fact, we have not experimented with different feeding practices, or with different quantities of Artemia. The focus of this work was to compare the breeding conditions in a non-natural environment with those in a natural environment in order to identify parameters to be considered in more specific subsequent experiments.
Comment 14: Results and Discussion. The authors should discuss the variations in fatty acid profiles between the two jellyfish populations in terms of stress responses under different environmental conditions.
Response 14: The synthesis, metabolism and catabolism of fatty acids is related to environmental stress in a complex way, in this work we have not focused on particular types of stress. Furthermore, we can infer that the breeding conditions are stressful for the holobiont C. andromeda, and we can argue that the main parameters are lighting, temperature and nutrition, but we have not done experiments aimed at identifying the effects on fatty acids or any other compounds, of each of these stressors.
Comment 15: Results and Discussion. What potential mechanisms could explain the observed enrichment of polyunsaturated fatty acids (PUFAs) during hydroalcoholic extraction, particularly in essential fatty acids such as arachidonic acid and docosahexaenoic acid?
Response 15: The hydroalcoholic extraction allows to extract nonpolar and partially polar molecules, the nonpolar molecules, such as fatty acids, can then be further separated from the hydroalcoholic extracted mixture by fractionation with acetonitrile according to our protocol. This determines a phase separation and an enrichment of the upper phase (UP) of lipids in particular of polyunsaturated fatty acids which are particularly preserved.
Comment 16: Results and Discussion. Given that the protein content is similar across the two populations, the authors should discuss the implications of the different protein distributions between the UP and LP phases in reared jellyfish compared to wild jellyfish.
Response 16: This was discussed in the section 2.3.3. paragraph at lines 523 – 533. Thanks to the reviewer's observation we have also added a sentence at lines 534-536.
Comment 17: Results and Discussion. Given the effect of artificial lighting on phenolic content, the authors should offer specific lighting adjustments to reduce stress and optimize the growth of reared C. andromeda
Response 17: One of the objectives of this work was also to identify key parameters for the growth and metabolism of C. andromeda in relation to their ability to biosynthesize bioactive compounds precisely by identifying some differences with natural populations that apparently live in optimal conditions. The next step is certainly to analyze each of the parameters in subsequent experiments.
Comment 18: Results and Discussion. The authors should explain more clearly what factors might contribute to the significant difference in peridinin concentrations between wild and farmed C. andromeda specimens.
Response 18: Peridinin (including its isomers) does not differ significantly between wild and farmed jellyfish (Table 4 and Table 7), the value indicates only individual variability.
Comment 19: Results and Discussion. Given the higher antioxidant activity in the total hydroalcoholic extract of reared jellyfish, the authors should propose some mechanisms that might underlie this observed increase.
Response 19: Thanks, we agreed with the reviewer and we modify the text at lines 610-613.
Comment 20: Methods. Section 3.2. The authors should indicate the sampling period for wild specimens.
Response 20: Thanks for alerting us; we added the missing test.
Comment 21: Methods. Section 3.11. Did the authors check the data for normality and homogeneity of variance prior to ANOVA?
Response 21: Prior to conducting the ANOVA we evaluated the data for normality and homogeneity of variance within GraphPad Prism. The results confirmed that the assumptions of ANOVA were met.

Reviewer 3 Report
Comments and Suggestions for Authors
In their manuscript, the authors describe a comparative study of the biochemical properties of jellyfish that were grown in an aquarium and caught in the sea. The work is a carefully executed classical biochemical study, and could well be published in the journal Marine Drugs, but after making some clarifications and adjustments.
1. Firstly, it is known that jellyfish are predators and have stinging cells that produce poison for defense and attack. Typically, these are polypeptides. What is known about the toxic components of those jellyfish. Has the composition of the venom been studied? Toxicity tests need to be included in the work.
2. Secondly, lines 279-297. Electrophoresis data is not enough to say what proteins these are. If these jellyfish have been studied before and there is data, then it is necessary to compare. If not, it is necessary to limit ourselves to masses and give a detailed description of other works where such proteins have been established or compare with genomic data... but this is worse. The comment is not correct, it should be replaced. Ideally, mass spectrometry is needed.
3. How can such huge measurement errors be explained? Could this be a measurement error? The data in Table 4 should be revised. The results for Diadinoxanthin (46.0 ± 45.5) and Peridinin 1 (81.8 ± 84.3) are alarming.
Minor comments
1. Line 89. w-3 and w-6 should be replaced by w3 and w6
2. Lines 182, 260, 309 There should be the periods at the end of the sentences.
3. Line 327. C. andromeda should be replaced by C. andromeda
The text may contain other errors, careful checking is recommended.
Author Response
Reviewer 3
The authors thank the reviewer for the time dedicated to the manuscript, and hope that the revised manuscript, also on the basis of the reviewer's comments, can be improved and useful for publication. Below our responses to the Reviewer’s comments.
Comment 1. Firstly, it is known that jellyfish are predators and have stinging cells that produce poison for defense and attack. Typically, these are polypeptides. What is known about the toxic components of those jellyfish. Has the composition of the venom been studied? Toxicity tests need to be included in the work.
Response 1. The jellyfish C. andromeda is considered to be moderately or slightly stinging and, in the (sub)tropical species of Cassiopea sp., novel autonomous stinging structures, called cassiosomes, have been identified, which are released together with mucus into the surrounding waters and are a major cause of non-contact stinging incidents in (sub)tropical coastal waters. Same structures could be present in other species of Cassiopea, although to our knowledge, there are no specific toxicology studies of the venom of C. andromeda present in the Mediterranean.
In this work we focused on the comparison between the cultured condition and the wild condition in which this aspect is certainly to be considered in a next set of experiments but we believe that it is not essential in this specific work. From literature and our data on C. andromeda and other jellyfish (De Domenico et al, De Rinaldis, et al ) it is known that the venom can be extracted and remain active in an aqueous extract (e.g. PBS) in non-denaturing conditions. The hydroalcoholic extraction in general, as well as the enzymatic hydrolysis eliminate the venom effect. Anyway, since jellyfish venoms are the main compounds with biological activity, it is certainly worth studying them in depth, especially if you intend to rear jellyfish.
Comment 2. Secondly, lines 279-297. Electrophoresis data is not enough to say what proteins these are. If these jellyfish have been studied before and there is data, then it is necessary to compare. If not, it is necessary to limit ourselves to masses and give a detailed description of other works where such proteins have been established or compare with genomic data... but this is worse. The comment is not correct, it should be replaced. Ideally, mass spectrometry is needed.
Response 2. Completely agree. Our working hypothesis was to compare only the protein patterns of denaturing electrophoretic analysis (SDS-PAGE) to highlight some differences both between the two jellyfish populations (wild and reared) and among the different fractions obtained in each sample able to support the other data of comparison reported here but also to give possible new information.
Since the number of microalgae in both samples is significant, and since there is a number of microalgae in the wild samples three times higher than those reared, we assumed that the differences could be related to the proteins of the symbiont or symbiosis metabolism. However, we have always discussed our data as hypotheses.
Comment 3. How can such huge measurement errors be explained? Could this be a measurement error? The data in Table 4 should be revised. The results for Diadinoxanthin (46.0 ± 45.5) and Peridinin 1 (81.8 ± 84.3) are alarming.
Response 3. Thanks for pointing out the typos. Corrected.
Minor comments
Comment 1a. Line 89. w-3 and w-6 should be replaced by w3 and w6ù
R.1 Thanks for reporting the typos, since Marine Drags format did not recognize the Greek character “ω”, we replaced them with omega-3 and omega-6.
Comment 2a. Lines 182, 260, 309 There should be the periods at the end of the sentences.
R2. Corrected. Thanks.
Comment 3a. Line 327. C. andromeda should be replaced by C. andromeda
R3. Corrected. Thanks.
The text may contain other errors, careful checking is recommended.
Thanks, we have proofread and corrected the errors in the text.

Round 2
Reviewer 3 Report
Comments and Suggestions for Authors
Accept in present form